# ODICE: Revealing the Mystery of Distribution Correction Estimation via Orthogonal-Gradient Update

**Liyuan Mao**[1*]   **Haoran Xu**[2*]   **Weinan Zhang**[1†]   **Xianyuan Zhan**[3,4†]

[1] Shanghai Jiao Tong University
[2] The University of Texas at Austin
[3] Institute for AI Industry Research (AIR), Tsinghua University
[4] Shanghai Artificial Intelligence Laboratory

`maoliyuan@sjtu.edu.cn, haoran.xu@utexas.edu`
`zhanxianyuan@air.tsinghua.edu.cn, wnzhang@sjtu.edu.cn`

## Abstract

In this study, we investigate the DIstribution Correction Estimation (DICE) methods, an important line of work in offline reinforcement learning (RL) and imitation learning (IL). DICE-based methods impose state-action-level behavior constraint, which is an ideal choice for offline learning. However, they typically perform much worse than current state-of-the-art (SOTA) methods that solely use action-level behavior constraint. After revisiting DICE-based methods, we find there exist two gradient terms when learning the value function using true-gradient update: forward gradient (taken on the current state) and backward gradient (taken on the next state). Using forward gradient bears a large similarity to many offline RL methods, and thus can be regarded as applying action-level constraint. However, directly adding the backward gradient may degenerate or cancel out its effect if these two gradients have conflicting directions. To resolve this issue, we propose a simple yet effective modification that projects the backward gradient onto the normal plane of the forward gradient, resulting in an orthogonal-gradient update, a new learning rule for DICE-based methods. We conduct thorough theoretical analyses and find that the projected backward gradient brings state-level behavior regularization, which reveals the mystery of DICE-based methods: the value learning objective does try to impose state-action-level constraint, but needs to be used in a corrected way. Through toy examples and extensive experiments on complex offline RL and IL tasks, we demonstrate that DICE-based methods using orthogonal-gradient updates (O-DICE) achieve SOTA performance and great robustness. Code is available at `https://github.com/maoliyuan/ODICE-Pytorch`.

## 1 Introduction

Offline Reinforcement Learning (RL) has attracted lots of attention as it enables learning policies by utilizing only pre-collected data without any costly or unsafe online interaction. It is a promising area for bringing RL into real-world domains, such as robotics (Kalashnikov et al., 2021), healthcare (Tang & Wiens, 2021), and industrial control (Zhan et al., 2022). However, offline RL is known to suffer from the value overestimation issue (Fujimoto et al., 2018): policy improvement entails querying the value function on values of actions produced by the learning policy, which are often unseen in the offline dataset. Those (potentially) out-of-distribution (OOD) actions can cause catastrophic extrapolation errors of the value function. To remedy this issue, current state-of-the-art (SOTA) offline RL algorithms typically impose different kinds of *action-level* behavior constraints into the learning objective, to limit how far the learning policy deviates from the behavior policy.

---

[*]Equal contribution, work done during internship at Institute for AI Industry Research, Tsinghua University.
[†]Corresponding authors.

Action-level constraint can be incorporated explicitly on the policy improvement step by calculating some divergence or distance metrics (Kumar et al., 2019; Wu et al., 2019; Nair et al., 2020; Fujimoto & Gu, 2021; Li et al., 2022a), or implicitly on the policy evaluation step by regularizing the learned value functions (Kumar et al., 2020; Kostrikov et al., 2021a; Xu et al., 2023; Garg et al., 2023).

However, action-level behavior constraint is not an ideal choice in offline RL. Although OOD states do not exist during training, they do occur at evaluation time. During evaluation, due to covariate shift and compounding errors, the policy could output undesired actions on encountered unseen states since it receives no learning signals on those OOD states during training. Hence, it is better to apply state and action distribution constraint simultaneously, *a.k.a.*, *state-action-level* behavior constraint, to force the policy to avoid or recover from both OOD states and actions. To the best of our knowledge, only a few works explicitly consider state-action-level behavior constraints (Li et al., 2022b; Zhang et al., 2022). These methods build on top of action-constrained offline RL algorithms, with additional steps to obtain and regularize OOD states by training dynamics models for OOD detection (Yang et al., 2021). Instead of manually adding the state-level constraint, there already exists an important line of work, DIstribution Correction Estimation (DICE) methods (Nachum & Dai, 2020), that is derived by constraining the joint state-action visitation distribution between the learning policy and behavior policy. By applying convex duality (Boyd et al., 2004), DICE-based methods elegantly transform the intractable state-action-level constraint into a tractable convex optimization objective that can be solved using only the offline dataset.

Although built upon the more ideal state-action-level behavior constraint, DICE-based methods are known to perform much worse than methods that only enforce action-level constraint (Fujimoto & Gu, 2021; Xu et al., 2023; Garg et al., 2023) on commonly-used benchmark datasets (Fu et al., 2020), which seems to be rather counterintuitive. In this work, we aim to resolve this issue and reveal the mystery of performance deficiencies in existing DICE-based methods. We first note that the derived value learning objectives of DICE-based methods contain the bellman residual term $(r + \gamma \mathbb{E}_{s'}[V(s')] - V(s))$, *i.e.*, value function should be learned using *true-gradient* update (Baird, 1995), which means the gradient of value function should be taken twice on two successor states $s$ and $s'$. We find that the forward gradient term (*i.e.,* the gradient taken on $s$) is actually imposing action-level constraint, which explains why recent work (Sikchi et al., 2023) that modifies DICE-based methods using *semi-gradient* update (Sutton et al., 1998) could match the performance of action-constrained offline RL methods. Furthermore, when diving deep into the backward gradient term (*i.e.,* the gradient taken on $s'$), we find that after a small modification, it is endowed with the ability to enforce state-level constraint. More specifically, we project the backward gradient onto the normal plane of the forward gradient, making these two gradient terms orthogonal. We term this method *orthogonal-gradient update*, by doing so, forward and backward gradients will not be canceled out by each other. Theoretically, orthogonal-gradient update has several appealing properties. It offers better convergence properties than semi-gradient update and makes the value function learn better representations across states, alleviating the feature co-adaptation issue (Kumar et al., 2022), while being more robust to state distributional shift (*i.e.*, OOD states).

We further propose a practical algorithm, O-DICE (Orthogonal-DICE), that incorporates V-DICE algorithm (Lee et al., 2021; Sikchi et al., 2023) with orthogonal-gradient update. We verify the effectiveness of O-DICE in both the offline RL setting and the offline Imitation Learning (IL) setting, where only one expert trajectory is given (Kostrikov et al., 2020; Zhang et al., 2023b). O-DICE not only surpasses strong action-constrained offline RL baselines, reaching SOTA performance in D4RL benchmarks (Fu et al., 2020), but also exhibits lower evaluation variances and stronger robustness to state distributional shift, especially on offline IL tasks, in which OOD states are more likely to occur, and being robust to them is crucial for high performance. To aid conceptual understanding of orthogonal-gradient update, we analyze its learned values in a simple toy setting, highlighting the advantage of its state-action-level constraint as opposed to true-gradient update or action-level constraint imposed by semi-gradient update. We also conduct validation experiments and validate the theoretical superiority of orthogonal-gradient updates.

## 2 MYSTERY OF DISTRIBUTION CORRECTION ESTIMATION (DICE)

In this section, we give a detailed derivation about how we reveal the mystery of DICE-based methods by using orthogonal-gradient update. We begin with some background introduction about differ-

ent types of DICE-based methods. We then dive deep into the gradient flow of DICE-based methods. We discuss the effect of two different parts in DICE's gradient flow and find a theory-to-practice gap. We thus propose the orthogonal-gradient update to fix this gap and instantiate a practical algorithm.

## 2.1 PRELIMINARIES

We consider the RL problem presented as a Markov decision process (Sutton et al., 1998), which is specified by a tuple $\mathcal{M}\langle \mathcal{S}, \mathcal{A}, \mathcal{P}, \rho_0, r, \gamma \rangle$ consisting of a state space, an action space, a transition probability function, a reward function, an initial state distribution, and the discount factor. The goal of RL is to find a policy $\pi(a|s)$ that maximizes expected return $\mathcal{J}(\pi) := \mathbb{E}[\sum_{t=0}^{\infty} \gamma^t \cdot r(s_t, a_t)]$, while $s_t, a_t$ satisfy $a_t \sim \pi(\cdot|s_t)$ and $s_{t+1} \sim \mathcal{P}(\cdot|s_t, a_t)$. $\mathcal{J}(\pi)$ has an equivalent dual form (Puterman, 2014) that can be written as $\mathcal{J}(\pi) = \mathbb{E}_{(s,a) \sim d^\pi}[r(s,a)]$, where $d^\pi(s,a) := (1-\gamma) \sum_{t=0}^{\infty} \gamma^t Pr(s_t = s, a_t = a)$ is the discounted state-action visitation distribution. In this work, we mainly focus on the offline setting, where learning from a static dataset $\mathcal{D} = \{s_i, a_i, r_i, s_i'\}_{i=1}^N$ is called offline RL and learning from a static dataset without reward labels $\mathcal{D} = \{s_i, a_i, s_i'\}_{i=1}^N$ is called offline IL. The dataset can be heterogenous and suboptimal, we denote the empirical behavior policy of $\mathcal{D}$ as $\pi_\mathcal{D}$, which represents the conditional distribution $p(a|s)$ observed in the dataset, and denote the empirical discounted state-action visitation distribution of $\pi_\mathcal{D}$ as $d^\mathcal{D}$.

DICE algorithms (Nachum & Dai, 2020) are motivated by bypassing unknown on-policy samples from $d^\pi$ in the dual form of $\mathcal{J}(\pi)$. They incorporate $\mathcal{J}(\pi)$ with a behavior constraint term $D_f(d^\pi || d^\mathcal{D}) = \mathbb{E}_{(s,a) \sim d^\mathcal{D}}[f(\frac{d^\pi(s,a)}{d^\mathcal{D}(s,a)})]$ where $f(x)$ is a convex function, *i.e.*, finding $\pi^*$ satisfying:

$$\pi^* = \arg\max_\pi \mathbb{E}_{(s,a) \sim d^\pi}[r(s,a)] - \alpha D_f(d^\pi || d^\mathcal{D}) \tag{1}$$

This objective is still intractable, however, by adding the *Bellman-flow* constraint (Manne, 1960), *i.e.*, $d(s,a) = (1-\gamma)d_0(s)\pi(a|s) + \gamma \sum_{(s',a')} d(s',a')p(s|s',a')\pi(a|s)$ (on state-action space) or $\sum_{a \in \mathcal{A}} d(s,a) = (1-\gamma)d_0(s) + \gamma \sum_{(s',a')} d(s',a')p(s|s',a')$ (on state space), we can get two corresponding dual forms which are tractable (by applying duality and convex conjugate, see Appendix D for details).

`Q-DICE`    $\max_\pi \min_Q (1-\gamma)\mathbb{E}_{s \sim d_0, a \sim \pi(\cdot|s)}[Q(s,a)] + \alpha\mathbb{E}_{(s,a) \sim d^\mathcal{D}}[f^*([\mathcal{T}^\pi Q(s,a) - Q(s,a)]/\alpha)]$

`V-DICE`    $\min_V (1-\gamma)\mathbb{E}_{s \sim d_0}[V(s)] + \alpha\mathbb{E}_{(s,a) \sim d^\mathcal{D}}[f^*([\mathcal{T}V(s,a) - V(s)]/\alpha)]$    (2)

where $f^*$ is the (variant of) convex conjugate of $f$. $d_0$ is the distribution of initial state $\rho_0$, to increase the diversity of samples, one often extends it to $d^\mathcal{D}$ by treating every state in a trajectory as initial state (Kostrikov et al., 2020). $\mathcal{T}^\pi Q(s,a) = r(s,a) + \gamma\mathbb{E}_{s' \sim p(\cdot|s,a), a' \sim \pi(\cdot|s')}[Q(s',a')]$ and $\mathcal{T}V(s,a) = r(s,a) + \gamma\mathbb{E}_{s' \sim p(\cdot|s,a)}[V(s')]$ represent the Bellman operator on $Q$ and $V$, respectively. In the offline setting, since $\mathcal{D}$ typically does not contain all possible transitions $(s,a,s')$, one actually uses an *empirical* Bellman operator that only backs up a single $s'$ sample, we denote the corresponding operator as $\hat{\mathcal{T}}^\pi$ and $\hat{\mathcal{T}}$.

Crucially, these two objectives rely only on access to samples from offline dataset $\mathcal{D}$. Note that in both cases for `Q-DICE` and `V-DICE`, the optimal solution is the same as their primal formulations due to strong duality. For simplicity, we consider `V-DICE` as an example for the rest derivation and discussion in our paper, note that similar derivation can be readily conducted for `Q-DICE` as well.

## 2.2 DECOMPOSE GRADIENT FLOW OF DICE

In the learning objective of `V-DICE`, we consider a parameterized value function $V$ with network parameters denoted as $\theta$. In objective (2), the effect of the linear term is clear that it pushes down the $V$-value of state samples equally. Whereas the effect of the nonlinear term which contains the Bellman residual term (*i.e.*, $\hat{\mathcal{T}}V(s,a) - V(s)$) is unclear because both $V(s')$ and $-V(s)$ would contribute to the gradient. We thus are interested in decomposing and investigating each part of the second term, the gradient flow could be formally written as:

$$\nabla_\theta f^*(\hat{\mathcal{T}}V(s,a) - V(s))) = (f^*)'(r + \gamma V(s') - V(s))(\gamma \underbrace{\nabla_\theta V(s')}_{\text{backward gradient}} - \underbrace{\nabla_\theta V(s)}_{\text{forward gradient}}) \tag{3}$$

where we consider $\alpha = 1$ for simplicity. Here we denote the two parts of the gradient as "forward gradient" (on $s$) and "backward gradient" (on $s'$) respectively, as shown in Eq.(3). A keen reader may note that Eq.(3) looks like algorithms in online RL that use *true-gradient* update (Baird, 1995). True-gradient algorithms are known to enjoy convergence guarantees to local minima under off-policy training with any function approximator. However, in online RL, most famous deep RL algorithms use *semi-gradient* update, *i.e.*, applying target network on $V(s')$ to freeze the backward gradient, such as DQN (Mnih et al., 2013), TD3 (Fujimoto et al., 2018) and SAC (Haarnoja et al., 2017). If we modify Eq.(3) to use semi-gradient update, we have

$$\nabla_\theta f^*(Q(s,a) - V(s))) = (f^*)'(Q(s,a) - V(s))(\gamma \underbrace{\nabla_\theta V(s')}_{\text{backward gradient}} - \underbrace{\nabla_\theta V(s)}_{\text{forward gradient}}) \qquad (4)$$

where $Q(s,a)$ denotes $\texttt{stop-gradient}(r + \gamma V(s'))$. Perhaps surprisingly, we find the result in Eq.(4) bears large similarity to the gradient flow of one recent SOTA offline RL algorithm, Exponential Q-Learning (EQL, Xu et al. (2023)), which imposes an implicit **action-level** behavior constraint (Reverse KL between $\pi$ and $\pi_\mathcal{D}$) with the following learning objective (with $\alpha$ in EQL equals to 1):

$$\min_V \ \mathbb{E}_{(s,a)\sim\mathcal{D}} \left[ \exp\left(Q(s,a) - V(s)\right) + V(s) \right] \qquad (5)$$

where $Q$ is learned to optimize $\mathbb{E}_{(s,a,s')\sim\mathcal{D}}[(r(s,a) + \gamma V(s') - Q(s,a))^2]$. If we take gradient of objective (5) with respect to $V$, we can get exactly Eq.(4) (with $f(x) = x \log x$)! This tells us that using the forward gradient alone is equal to imposing only action-level constraint. Our finding is consistent with one recent work (Sikchi et al., 2023) that finds $\texttt{V-DICE}$ using semi-gradient update is doing implicit policy improvement to find the best action in the dataset $\mathcal{D}$.

EQL has shown superior performance on D4RL benchmark tasks, however, OptiDICE (Lee et al., 2021), whose learning objective is exactly Eq.(2), shows significantly worse performance compared to EQL. It's obvious that the only difference between semi-DICE algorithm (EQL) and true-DICE algorithm (OptiDICE) is the backward gradient. Then one gap between theory and practice occurs: although the backward gradient term is considered to be necessary from the theoretical derivation, the practical result reveals that it is harmful and unnecessary.

### 2.3 Fix The Gap by Orthogonal-gradient Update

As said, the forward gradient is crucial for implicit maximization to find the best action. However, adding the backward gradient will probably interfere with it because a state $s$ and its successor state $s'$ are often similar under function approximation. The backward gradient may cancel out the effect of the forward gradient, leading to a catastrophic unlearning phenomenon. To remedy this, we adopt a simple idea that projects the backward gradient onto the normal plane of the forward gradient, making these two gradient terms orthogonal. The projected backward gradient can be written as

$$\nabla_\theta^\perp V(s') = \nabla_\theta V(s') - \frac{\nabla_\theta V(s)^\top \nabla_\theta V(s')}{\|\nabla_\theta V(s)\|^2} \nabla_\theta V(s)$$

Intuitively, the projected backward gradient will not interfere with the forward gradient while still retaining information from the backward gradient. We will further show in the next section that the projected backward gradient is actually doing state-level constraint. After getting the projected backward gradient, we add it to the forward gradient term, resulting in a new gradient flow,

$$\nabla_\theta f^* \left( \hat{\mathcal{T}} V(s,a) - V(s) \right) := (f^*)'(r + \gamma V(s') - V(s)) \left( \gamma \cdot \eta \nabla_\theta^\perp V(s') - \nabla_\theta V(s) \right)$$

where we use one hyperparameter $\eta > 0$ to control the strength of the projected backward gradient against the forward gradient. We call this new update rule as *orthogonal-gradient* update and give the illustration of it in Figure 1. We can see that the orthogonal-gradient can be balanced between true-gradient and semi-gradient by choosing different $\eta$.

After adding the gradient of the first linear term in objective (2), we can get the new full gradient flow of learning (2). We then propose a new offline algorithm, O-DICE (Orthogonal-DICE), that incorporates $\texttt{V-DICE}$ algorithm with orthogonal-gradient update. The policy extraction part in O-DICE is the same as $\texttt{V-DICE}$ by using weighted behavior cloning $\max_\pi \mathbb{E}_{(s,a)\sim\mathcal{D}}[\omega^*(s,a) \log \pi(a|s)]$, where $\omega^*(s,a) := \frac{d^{\pi^*}(s,a)}{d^\mathcal{D}(s,a)} = \max(0, (f')^{-1}(\hat{\mathcal{T}} V^*(s,a) - V^*(s)))$.

**Practical Consideration**    We now describe some practical considerations in O-DICE. O-DICE could be implemented with just a few adjustments to previous true/semi-DICE algorithms. To compute the forward and backward gradient, we apply target network to $V(s')$ when computing the forward gradient and apply target network to $V(s)$ when computing the backward gradient, *a.k.a*, the *bidirectional target network* trick (Zhang et al., 2019b). We also apply one trick from Sikchi et al. (2023) that rewrites objective (2) from $\alpha$ to $\lambda$ as

$$\min_V \mathbb{E}\left[(1-\lambda)V(s) + \lambda f^*(\mathcal{T}V(s,a) - V(s))\right]$$

where $\lambda \in (0,1)$ trades off linearly between the first term and the second term. This trick makes hyperparameter tuning easier as $\alpha$ has a nonlinear dependence through the function $f$. Analogous to to previous DICE algorithms, we alternate the updates of $V$ and $\pi$ until convergence although the training of them are decoupled. Note that different from almost all previous offline RL algorithms, we find that O-DICE does not need to use the double Q-learning trick (Fujimoto et al., 2018), which suggests that using orthogonal-gradient update regularizes the value function better and greatly enhances training stability. We summarize the pseudo-code of O-DICE in Algorithm 1.

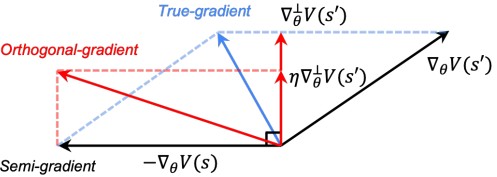

Figure 1: Illustration of orthogonal-gradient update

---

**Algorithm 1** Orthogonal DICE

---

**Require:** $\mathcal{D}, \lambda, \eta$, learning rate $\alpha$
1: Initialize $V_\theta, V_{\bar\theta}, \pi_\phi$
2: **for** $t = 1, 2, \cdots, N$ **do**
3:     Sample transitions $(s, a, r, s') \sim \mathcal{D}$
4:     Calculate forward gradient
        $\overrightarrow{g_\theta} = \nabla_\theta f^*(r + \gamma V_{\bar\theta}(s') - V_\theta(s))$
5:     Calculate backward gradient
        $\overleftarrow{g_\theta} = \nabla_\theta f^*(r + \gamma V_\theta(s') - V_{\bar\theta}(s))$
6:     Calculate projected backward gradient
$$g_\theta^\perp = \overleftarrow{g_\theta} - \frac{(\overleftarrow{g_\theta})^\top \overrightarrow{g_\theta}}{\|\overrightarrow{g_\theta}\|^2}\overrightarrow{g_\theta}$$
7:     Update $\theta$ by
$$\theta \leftarrow \theta - \alpha\left((1-\lambda)\nabla_\theta V_\theta(s) + \lambda(\overrightarrow{g_\theta} + \eta g_\theta^\perp)\right)$$
8:     Calculate BC weight
    $\omega(s,a) = \max\left(0, (f')^{-1}\left(r + \gamma V_\theta(s') - V_\theta(s)\right)\right)$
9:     Update $\pi_\phi$ by
        $\max_\phi\left[\omega(s,a)\log\pi_\phi(a|s)\right]$
10:     Update $\bar\theta$ with exponential moving average
        $\bar\theta \leftarrow \tau\theta + (1-\tau)\bar\theta$
11: **end for**

---

# 3    ANALYSIS

Although our method is simple and straightforward, in this section, we give a thorough theoretical analysis and use a didactic toy example to show several benefits of using orthogonal-gradient update.

## 3.1    THEORETIC RESULTS

We begin by introducing the notation used in our analysis. We denote the nonlinear loss in O-DICE, $f^*(r + \gamma V(s') - V(s))$ as $\mathcal{L}^\theta(s, s')$. We use $L_1^\theta(s)$ to denote $f^*(r + \gamma \overline{V}(s') - V(s))$, whose gradient corresponds to the forward gradient (multiply a value). We use $L_2^\theta(s')$ to denote $f^*(r + \gamma V(s') - \overline{V}(s))$, whose gradient corresponds to the backward gradient (multiply a value). We define the projected part and the remaining parallel part of $\nabla_\theta L_2^\theta(s')$ with respect to $\nabla_\theta L_1^\theta(s)$ as $\nabla_\theta^\perp L_2^\theta(s')$ and $\nabla_\theta^\| L_2^\theta(s')$, respectively. It's easy to know that we can decompose the orthogonal-gradient update in O-DICE into two steps, which are

$$\theta' = \theta - \alpha\nabla_\theta L_1^\theta(s)$$
$$\theta'' = \theta' - \alpha\nabla_\theta^\perp L_2^\theta(s')$$

where $\alpha$ is the learning rate. We first introduce two theoretical benefits of using the orthogonal-gradient update from the optimization perspective.

**Theorem 1.** *In orthogonal-gradient update, $L_1^{\theta''}(s) - L_1^{\theta'}(s) = 0$ (under first order approximation).*

This theorem formally says the interference-free property in orthogonal-gradient update, that is, the gradient change in loss $L_2(s')$ won't affect loss $L_1(s)$. This means that orthogonal-gradient update preserves the benefit of semi-gradient update while still making loss $L_2(s')$ decrease. Note that under general setting, $L_1^{\theta''}(s) - L_1^{\theta'}(s) = 0$ holds only by orthogonal-gradient update, any other gradient modification techniques (Baird, 1995; Zhang et al., 2019b; Durugkar & Stone, 2018) that keep part of $\nabla_\theta^\| L_2^\theta(s')$ will make $L_1^{\theta''}(s) - L_1^{\theta'}(s) \neq 0$.

**Theorem 2.** *Define the angle between $\nabla_\theta V(s)$ and $\nabla_\theta V(s')$ as $\phi(s, s')$. In orthogonal-gradient update, if $\phi(s, s') \neq 0$, then for all $\eta > \frac{1}{\sin^2 \phi(s,s')} \left( \cos \phi(s, s') \frac{\|\nabla_\theta V(s)\|}{\gamma \|\nabla_\theta V(s')\|} - (\frac{\|\nabla_\theta V(s)\|}{\gamma \|\nabla_\theta V(s')\|})^2 \right)$, we have $\mathcal{L}^{\theta''}(s, s') - \mathcal{L}^{\theta}(s, s') < 0$.*

This theorem gives a convergence property that using orthogonal-gradient update could make the overall loss objective monotonically decrease. Note that similar convergence guarantee occurs only in true-gradient update, while semi-gradient update is known to suffer from divergence issue (Zhang et al., 2019b). Actually, to have a convergence guarantee, we only need to keep the gradient within the same half-plane that $\nabla_\theta \mathcal{L}^\theta(s, s')$ lies in. This means that we can adjust $\eta$ to make the orthogonal gradient keep in the same half-plane of the true-gradient while keeping close to the semi-gradient if needed, as shown in this theorem. Note that $s$ and $s'$ are often similar, so the value of $\|\nabla_\theta V(s)\|$ and $\|\nabla_\theta V(s')\|$ are close to each other and the ratio $\frac{\|\nabla_\theta V(s)\|}{\gamma \|\nabla_\theta V(s')\|}$ will probably $> 1$, which indicates that the choice of $\eta$ may be fairly easy to satisfy this theorem (*e.g.*, $\eta > 0$).

From Theorem 1 and Theorem 2 we know that orthogonal-gradient update could make loss $\mathcal{L}(s, s')$ and $L_1(s)$ decrease simultaneously, so as to share the best of both worlds: not only does it preserve the action-level constraint in semi-gradient update, but also owns the convergence guarantee in true-gradient update. We now further analyze the influence of orthogonal-gradient update, we show in the following theorem that it has a deep connection with the feature dot product of consecutive states in value function ($\Psi_\theta(s, s') = \nabla_\theta V(s)^\top \nabla_\theta V(s')$).

**Theorem 3** (Orthogonal-gradient update helps alleviate feature co-adaptation)**.** *Assume the norm of $\nabla_\theta V(s)$ is bounded, i.e. $\forall s, m \leq \|\nabla_\theta V(s)\|^2 \leq M$. Define consecutive value curvature $\xi(\theta, s, s') = \nabla_\theta^\perp V(s')^\top H(s) \nabla_\theta^\perp V(s')$, where $H(s)$ is the Hessian matrix of $V(s)$. Assume $\xi(\theta, s, s') \geq l \cdot \|\nabla_\theta^\perp V(s')\|^2$ where $l > 0$. Then in orthogonal-gradient update, we have*

$$\Psi_{\theta''}(s, s') - \Psi_{\theta'}(s, s') \leq -\alpha\eta\gamma \cdot (f^*)'(r + \gamma V(s') - V(s))[\sin^2 \phi(s, s') \cdot l \cdot m + \beta \cdot M] \quad (6)$$

*where $\beta$ is a constant close to 0 if the condition number of $H(s)$ is small (Cheney & Kincaid, 2012).*

Feature co-adaptation, where $\Psi_\theta(s, s')$ is large so that features of different states learned by the value function become very similar, is known to be a severe issue in offline deep RL (Kumar et al., 2021). While this theorem formally says that after using orthogonal-gradient update, the difference of $\Psi_\theta(s, s')$ could be bounded. In RHS of Eq.(6), $\beta$ is a constant that captures the difference between minimal and maximal eigenvalue in $H(s)$. $\beta$ will be close to 0 if neural network $V(s)$ is numerically well-behaved (*i.e.*, the condition number of $H(s)$ is small), which can be achieved by using some practical training tricks such as orthogonal regularization (Brock et al., 2016) or gradient penalty (Gulrajani et al., 2017) during training $V$. Also, note that $(f^*)'(x) \geq 0$ always holds because $f^*$ is chosen to be monotonically increasing (see proof in Appendix B for details, we highlight that this **only** holds in DICE-based methods, minimizing Bellman equation such that $f^* = x^2$ won't let it hold). Hence, Eq.(6) $< 0$ will probably hold, this means that using orthogonal-gradient update could help alleviate feature co-adaptation issue, learning better representations for different states. We show in the following theorem that better representation brings better state-level robustness of $V(s)$, akin to imposing the state-level behavior constraint.

**Theorem 4** (How feature co-adaptation affects state-level robustness)**.** *Under linear setting when $V(s)$ can be expressed as $\nabla_\theta V(s)^\top \theta$, assume $\|\nabla_\theta V(s)\|^2 \leq M$, then there exists some small noise $\varepsilon \in \mathbb{R}^{|\mathcal{S}|}$ such that $V(s' + \varepsilon) - V(s)$ will have different sign with $V(s') - V(s)$, if $\Psi_\theta(s, s') > M - C \cdot \|\varepsilon\|^2$ for some constant $C > 0$.*

This theorem says that if feature co-adaptation happens, *i.e.*, $\Psi_\theta(s, s')$ is larger than a threshold, a small noise adding to $V(s')$ will largely change its value, or the learning signal for policy. If the sign (or value) of $V(s') - V(s)$ can be easily flipped by some noise $\varepsilon$, not only the policy extraction part during training will be affected (as two similar $s'$ will have totally different BC weight), but also the optimal policy induced from $V$ is likely to be misled by OOD states encountered during evaluation and take wrong actions. As shown in Theorem 3, using orthogonal-gradient update helps alleviate feature co-adaptation and enables $V$ to learn better representations to distinguish different states, thus being more robust to OOD states.

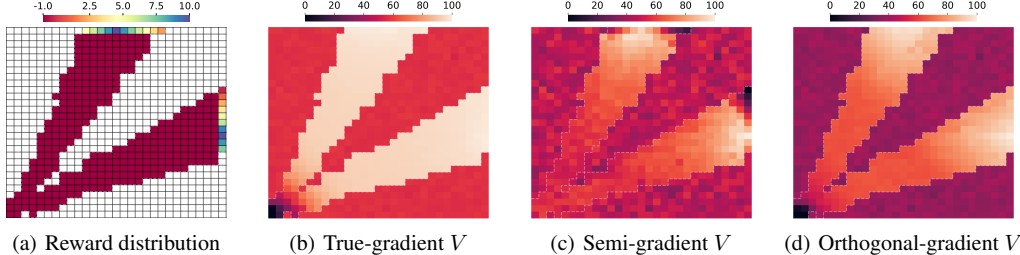

| (a) Reward distribution | (b) True-gradient $V$ | (c) Semi-gradient $V$ | (d) Orthogonal-gradient $V$ |

Figure 2: Visualizations of the value function $V$ learned by using different updating rules in a grid world toycase, the base algorithm is V-DICE. We normalize the value of $V$ to $[0, 100]$ for fair comparison. The borderline of the offline dataset support is marked by white dashed lines. Orthogonal-gradient update better distinguishes both different in-distribution actions and in-distribution states vs. OOD states.

## 3.2 A Toy Example

We now use a toy example to validate the above-mentioned benefits of using orthogonal-gradient update. We consider a $30 \times 30$ grid world navigation task where the agent starts from the bottom-left corner $(0, 0)$ and the goal is to reach a given area at the top ($(10, 30)$ to $(20, 30)$) or right border ($(30, 10)$ to $(30, 20)$). The action space contains 4 discrete actions: up, down, left, and right. Every time the agent chooses an action, it will move towards that direction for one grid, receiving a $-1$ reward, unless it reaches the goal area, which will give a different, positive reward, as shown in Figure 2(a). The offline dataset is collected by running a goal-reaching policy, to add some stochasticity, with a probability of 0.05 the output action will be shifted to a random action. We use white dashed lines in Figure 2 to plot the borderlines of the support of the collected dataset.

We run V-DICE with semi-gradient, true-gradient, and orthogonal-gradient, and visualize the learned $V$-value (with normalization) in Figure 2(b)-(d). The corresponding policy can be extracted implicitly by choosing a reachable state (grid) with the highest $V$-value, at a given state (grid). It can be shown from the result that orthogonal-gradient update could learn a value function (or policy) that successfully finds a path from the start to the desired location, indicating the effect of imposing action-level constraint. Meanwhile, it successfully distinguishes between in-distribution states and OOD states, the values of OOD states are much lower than that of in-distribution states, which reveals better feature of the value function is learned by imposing state-level constraint. Using semi-gradient update could learn a good policy to find the best action but lose OOD state robustness (the color of OOD states and in-distribution states are similar, once the agent gets into OOD states, it can hardly come back), while true-gradient update could not even learn a good policy because of the conflicting gradient issue.

From this toycase, we can clearly observe that using orthogonal-gradient update brings state-action-level constraint, it helps improve the robustness of OOD states while still preserving the ability to find the best dataset actions.

## 4 Experiments

In this section, we present empirical evaluations of O-DICE. We first evaluate O-DICE against other baseline algorithms on benchmark offline RL datasets. We then conduct more in-depth investigations on the policy robustness of O-DICE and other algorithms. Lastly, we evaluate O-DICE on offline IL tasks to demonstrate its superior performance. Hyperparameters and experimental details are listed in Appendix D.

### 4.1 Results on D4RL Benchmarks

We first evaluate O-DICE on the D4RL benchmark (Fu et al., 2020), including MuJoCo locomotion tasks and AntMaze navigation tasks. We compare O-DICE with previous SOTA baselines that only enforce action-level constraint, including TD3+BC (Fujimoto & Gu, 2021), CQL (Kumar et al., 2020), IQL (Kostrikov et al., 2021b), and EQL (Xu et al., 2023). We also compare O-DICE with

Table 1: Averaged normalized scores of O-DICE against other baselines. The scores are taken over the final 10 evaluations with 5 seeds. O-DICE achieves the highest score in 13 out of 15 tasks, outperforming previous SOTA offline RL methods and other DICE-based methods.

| D4RL Dataset | constrain $\pi(a\|s)$ | | | | constrain $d^\pi(s,a)$ | | | |
|---|---|---|---|---|---|---|---|---|
| | TD3+BC | CQL | IQL | EQL | OptiDICE | $f$-DVL | S-DICE | O-DICE |
| halfcheetah-m | 48.3 | 44.0 | 47.4 ±0.2 | 47.2 ±0.2 | 45.8 ±0.4 | 47.7 | 47.3±0.2 | 47.4±0.2 |
| hopper-m | 59.3 | 58.5 | 66.3 ±5.7 | 74.6 ±3.4 | 46.4 ±3.9 | 63.0 | 61.5±2.3 | 86.1 ±4.0 |
| walker2d-m | 83.7 | 72.5 | 72.5 ±8.7 | 83.2 ±4.6 | 68.1 ±4.5 | 80.0 | 78.6±7.1 | 84.9 ±2.3 |
| halfcheetah-m-r | 44.6 | 45.5 | 44.2 ±1.2 | 44.5 ±0.7 | 31.7 ±0.8 | 42.9 | 40.6±2.6 | 44.0±0.3 |
| hopper-m-r | 60.9 | 95.0 | 95.2 ±8.6 | 98.1 ±3.3 | 20.0 ±6.6 | 90.7 | 64.6±19.4 | 99.9 ±2.7 |
| walker2d-m-r | 81.8 | 77.2 | 76.1 ±7.3 | 76.6 ±3.8 | 17.9±5.9 | 52.1 | 37.3±5.1 | 83.6 ±4.1 |
| halfcheetah-m-e | 90.7 | 90.7 | 86.7 ±5.3 | 90.6 ±0.4 | 59.7 ±3.0 | 89.3 | 92.8±0.7 | 93.2 ±0.6 |
| hopper-m-e | 98.0 | 105.4 | 101.5 ±7.3 | 105.5 ±2.2 | 51.3 ±3.5 | 105.8 | 108.4±1.9 | 110.8 ±0.6 |
| walker2d-m-e | 110.1 | 109.6 | 110.6 ±1.0 | 110.2 ±0.8 | 104.0 ±5.1 | 110.1 | 110.1±0.2 | 110.8 ±0.2 |
| antmaze-u | 78.6 | 84.8 | 85.5 ±1.9 | 93.2 ±1.4 | 56.0 ±2.8 | 83.7 | 88.6±2.5 | 94.1 ±1.6 |
| antmaze-u-d | 71.4 | 43.4 | 66.7 ±4.0 | 65.0 ±2.3 | 48.6 ±7.4 | 50.4 | 69.3±5.0 | 79.5 ±3.3 |
| antmaze-m-p | 10.6 | 65.2 | 72.2 ±5.3 | 77.5 ±3.7 | 0.0 ±0.0 | 56.7 | 50.6±11.1 | 86.0 ±1.6 |
| antmaze-m-d | 3.0 | 54.0 | 71.0 ±3.2 | 70.0 ±4.2 | 0.0 ±0.0 | 48.2 | 66.0±3.6 | 82.7 ±4.9 |
| antmaze-l-p | 0.2 | 38.4 | 39.6 ±4.5 | 45.6 ±4.8 | 0.0 ±0.0 | 36.0 | 34.0±9.0 | 55.9 ±3.9 |
| antmaze-l-d | 0.0 | 31.6 | 47.5 ±4.4 | 42.5 ±5.2 | 0.0 ±0.0 | 44.5 | 32.6±4.1 | 54.0 ±4.8 |

other DICE-based algorithms that intend to do state-action-level constraint, including OptiDICE (Lee et al., 2021), $f$-DVL (Sikchi et al., 2023) and our implementation of semi-DICE (S-DICE) which updates V-DICE using semi-gradient update.

The result shows that O-DICE outperforms all previous SOTA algorithms on almost all MuJoCo and AntMaze tasks. This suggests that O-DICE can effectively learn a strong policy from the dataset, even in challenging AntMaze tasks that require learning a high-quality value function so as to "stitch" good trajectories. The large performance gap between O-DICE (using orthogonal-gradient update) and OptiDICE (using true-gradient update) demonstrates the necessity of using the projected backward gradient. Moreover, O-DICE shows consistently better performance over EQL, $f$-DVL, and S-DICE (using semi-gradient update) on almost all datasets, especially on sub-optimal datasets that contain less or few near-optimal trajectories. This indicates the superiority of state-action-level constraint over only action-level constraint, by learning a value function with better representation to successfully distinguish good trajectories (actions) from bad ones.

## 4.2 RESULTS ON POLICY ROBUSTNESS

The aggregated mean score in D4RL can't fully reflect the robustness of an algorithm because, even if the average performance is reasonable, the agent may still perform poorly on some episodes. Here we compare O-DICE with S-DICE and EQL on the percent difference of the worst episode during the 10 evaluation episodes at the last evaluation (Fujimoto & Gu, 2021), we also plot the average value of feature dot product, $\nabla_\theta V(s)^\top \nabla_\theta V(s')$, as shown in Figure 3. It's obvious that a robust policy should achieve consistently high per-

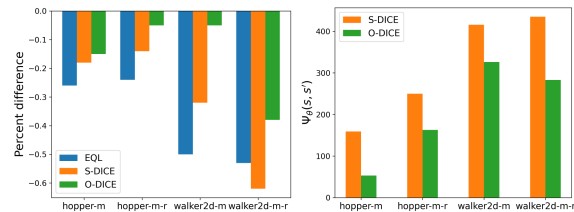

Figure 3: Left: percent difference of the worst episode during the 10 evaluation episodes at the last evaluation of different offline RL algorithms. Right: mean value of $\nabla_\theta V(s)^\top \nabla_\theta V(s')$ over the dataset of S-DICE and O-DICE.

formance over multiple evaluations, i.e., having a small percent difference. Moreover, Theorem 4 shows that a low value of $\nabla_\theta V(s)^\top \nabla_\theta V(s')$ is necessary for robustness. It can be observed that O-DICE achieves much smaller percent differences than action-constrained offline RL methods like EQL and S-DICE, which empirically verifies Theorem 4. The significantly decreased feature dot-product value $\nabla_\theta V(s)^\top \nabla_\theta V(s')$ after applying orthogonal-gradient also agrees with Theorem 3.

### 4.3 RESULTS ON OFFLINE IMITATION LEARNING

We then evaluate O-DICE in the offline imitation learning setting, where only one expert trajectory is given (Kostrikov et al., 2020). In this setting, OOD states are fairly easy to occur, and being robust to them is crucial for high performance (Rajaraman et al., 2020). We compare O-DICE with BC (Pomerleau, 1989), ValueDICE (Kostrikov et al., 2020) and IQLearn Garg et al. (2021). ValueDICE and IQLearn are two Q-DICE algorithms that use true-gradient

Table 2: Normalized return of BC, ValueDICE, IQLearn and O-DICE on offline IL tasks.

| Dataset | | BC | ValueDICE | IQLearn | O-DICE |
|---|---|---|---|---|---|
| Hopper-expert | Traj 1 | 56.0 | 33.8 | 75.1 | **78.8** |
| | Traj 2 | 36.2 | 23.5 | **56.2** | 47.8 |
| | Traj 3 | 50.5 | 24.7 | 47.1 | **55.1** |
| | Traj 4 | 87.2 | 90.4 | 94.2 | **98.5** |
| Walker2d-expert | Traj 1 | 64.6 | 23.6 | 50.2 | **75.6** |
| | Traj 2 | 43.3 | 17.1 | 40.5 | **65.8** |
| | Traj 3 | 38.4 | 60.6 | **66.7** | 62.7 |
| | Traj 4 | 33.0 | 40.5 | 44.2 | **61.6** |

update and semi-gradient update, respectively. As we find performance varies a lot by using different trajectories, we select four trajectories from D4RL expert datasets, each containing 1000 transitions. It can be shown in Table 2 that although all algorithms can't match the expert performance due to extremely limited data, O-DICE consistently performs better than all other baselines. This justifies the benefits of using orthogonal-gradient update, it enables better state-level robustness against OOD states by learning better feature representation.

## 5 RELATED WORK

**Offline RL** To tackle the distributional shift problem, most model-free offline RL methods augment existing off-policy RL methods with an action-level behavior regularization term. Action-level regularization can appear explicitly as divergence penalties (Wu et al., 2019; Kumar et al., 2019; Xu et al., 2021; Fujimoto & Gu, 2021; Cheng et al., 2023; Li et al., 2023b), implicitly through weighted behavior cloning (Wang et al., 2020; Nair et al., 2020), or more directly through careful parameterization of the policy (Fujimoto et al., 2019; Zhou et al., 2020). Another way to apply action-level regularization is via modification of value learning objective to incorporate some form of regularization, to encourage staying near the behavioral distribution and being pessimistic about OOD state-action pairs (Kumar et al., 2020; Kostrikov et al., 2021a; Xu et al., 2022c; 2023; Wang et al., 2023; Niu et al., 2022). There are also several works incorporating action-level regularization through the use of uncertainty (An et al., 2021; Bai et al., 2021) or distance function (Li et al., 2022a). Another line of methods, on the contrary, imposes action-level regularization by performing some kind of imitation learning on the dataset. When the dataset is good enough or contains high-performing trajectories, we can simply clone or filter dataset actions to extract useful transitions (Xu et al., 2022b; Chen et al., 2020; Zhang et al., 2023a; Zheng et al., 2024), or directly filter individual transitions based on how advantageous they could be under the behavior policy and then clones them (Brandfonbrener et al., 2021; Xu et al., 2022a). There are also some attempts to include state-action-level behavior regularization (Li et al., 2022b; Zhang et al., 2022), however, they typically require costly extra steps of model-based OOD state detection (Li et al., 2022b; Zhang et al., 2022).

**DICE-based Methods** DICE-based methods perform stationary distribution estimation, and many of them have been proposed for off-policy evaluation: DualDICE (Nachum et al., 2019a), GenDICE (Zhang et al., 2019a), GradientDICE (Zhang et al., 2020). Other lines of works consider reinforcement learning: AlgaeDICE (Nachum et al., 2019b), OptiDICE (Lee et al., 2021), CoptiDICE (Lee et al., 2022), $f$-DVL (Sikchi et al., 2023); offline policy selection: (Yang et al., 2020); offline imitation learning: ValueDICE (Kostrikov et al., 2020), OPOLO (Zhu et al., 2020), IQlearn (Garg et al., 2021), DemoDICE (Kim et al., 2021), SmoDICE (Ma et al., 2022); reward learning: RGM (Li et al., 2023a) All of these DICE methods are either using true-gradient update or semi-gradient update, while our paper provides a new update rule: orthogonal-gradient update.

## 6 CONCLUSIONS

In this paper, we revisited DICE-based method. We provide a new understanding of DICE-based method and successfully unravel its mystery: the value learning objective in DICE does try to impose state-action-level constraint, but needs to be used in a corrected way, that is orthogonal-gradient update. By doing so, DICE-based methods enjoy not only strong theoretical guarantees, but also favorable empirical performance. It achieves SOTA results on both offline RL and offline IL tasks.

## ACKNOWLEDGEMENT

This work is supported by National Key Research and Development Program of China under Grant (2022YFB2502904). The SJTU team is partially supported by Shanghai Municipal Science and Technology Major Project (2021SHZDZX0102) and National Natural Science Foundation of China (62322603, 62076161). We thank Amy Zhang and Bo Dai for insightful discussions.

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

## A   A RECAP OF DISTRIBUTION CORRECTION ESTIMATION

For the sake of completeness, we give a more detailed review of DICE in this section. We won't cover all variants of DICE algorithms but only focus on how to derive the final optimization objective. For a more thorough discussion, please refer to (Nachum & Dai, 2020) and (Sikchi et al., 2023).

### A.1   CONVEX CONJUGATE AND $f$-DIVERGENCE

Given a convex function $f : \mathbb{R} \to \mathbb{R}$, it's convex conjugate is defined by:

$$f^*(y) = \sup\{y \cdot x - f(x), x \in \mathbb{R}\} \tag{7}$$

Sometimes a proper choice of $f$ can make $f^*$ have a close-form solution. For example, $f(x) = (x-1)^2$ can induce $f^*(y) = y(\frac{y}{4} + 1)$.

Moving forward, we'll introduce $f$-divergence. Given two probability distributions over $\Omega$: $P$ and $Q$, if $P$ is absolutely continuous with respect to $Q$, the $f$-divergence from $P$ to $Q$ is defined as:

$$D_f(P||Q) = \mathbb{E}_{\omega \in Q}\left[f\left(\frac{P(\omega)}{Q(\omega)}\right)\right] \tag{8}$$

### A.2   DISTRIBUTION CORRECTION ESTIMATION

DICE algorithms consider RL problems as convex programming problems with linear constraints and then apply Fenchel-Rockfeller duality or Lagrangian duality to transform the problem into a constraint-free formulation (Nachum & Dai, 2020; Lee et al., 2021). More specifically, consider the following regularized RL problem:

$$\max_{\pi} \mathbb{E}_{(s,a) \sim d^\pi}[r(s,a)] - \alpha D_f(d^\pi(s,a)||d^{\mathcal{D}}(s,a)) \tag{9}$$

Here $D_f(d^\pi(s,a)||d^{\mathcal{D}}(s,a))$ is a regularizer who encourages the agent to stay close to data support from the perspective of $(s,a)$ occupancy measure. Directly solving $\pi^*$ is impossible because it's intractable to calculate $d^\pi(s,a)$. However, as shown in (Nachum & Dai, 2020; Lee et al., 2021), one can consider $d^\pi$ as another variable $d$ in the optimization problem which satisfies Bellman-flow constraints. Then we obtain an optimization problem that's simultaneously about $d$ and $\pi$:

$$\max_{\pi,d} \mathbb{E}_{(s,a) \sim d}[r(s,a)] - \alpha D_f(d(s,a)||d^{\mathcal{D}}(s,a))$$
$$\text{s.t. } d(s,a) = (1-\gamma)d_0(s)\pi(a|s) + \gamma \sum_{(s',a')} d(s',a')p(s|s',a')\pi(a|s), \forall s \in \mathcal{S}, a \in \mathcal{A} \tag{10}$$

As is pointed out in (Nachum & Dai, 2020), the original Bellman-flow over-constrains the problem. By summing constraints over $a$, we can eliminate $\pi(a|s)$ in the constraints and get another problem solely depend on $d$:

$$\max_{d \geq 0} \mathbb{E}_{(s,a) \sim d}[r(s,a)] - \alpha D_f(d(s,a)||d^{\mathcal{D}}(s,a))$$
$$\text{s.t. } \sum_{a \in \mathcal{A}} d(s,a) = (1-\gamma)d_0(s) + \gamma \sum_{(s',a')} d(s',a')p(s|s',a'), \forall s \in \mathcal{S} \tag{11}$$

Note that in this problem we have to set the feasible region to be $\{d : \forall s \in \mathcal{S}, a \in \mathcal{A}, d(s,a) \geq 0\}$ because we can't rely on the non-negativity of $\pi$ to force $d \geq 0$. Applying Lagrangian duality, we

can get the following optimization target following (Lee et al., 2021):

$$\min_{V(s)} \max_{d \geq 0} \mathbb{E}_{(s,a)\sim d}[r(s,a)] - \alpha D_f(d(s,a)||d^{\mathcal{D}}(s,a))$$

$$+ \sum_s V(s)\Big((1-\gamma)d_0(s) + \gamma \sum_{(s',a')} d(s',a')p(s|s',a') - \sum_a d(s,a)\Big)$$

$$= \min_{V(s)} \max_{d \geq 0} (1-\gamma)d_0(s)\sum_s V(s) + \mathbb{E}_{(s,a)\sim d}[r(s,a)]$$

$$+ \underbrace{\sum_s V(s) \sum_{(s',a')} d(s',a')p(s|s',a') - \sum_{s,a} d(s,a)V(s) - \alpha D_f(d(s,a)||d^{\mathcal{D}}(s,a))}_{(*)}$$

$$= \min_{V(s)} \max_{\omega \geq 0} (1-\gamma)\mathbb{E}_{d_0(s)}[V(s)]$$

$$+ \mathbb{E}_{s,a\sim d^{\mathcal{D}}}\Big[\omega(s,a)\Big(r(s,a) + \gamma \sum_{s'} p(s'|s,a)V(s') - V(s)\Big)\Big] - \mathbb{E}_{s,a\sim d^{\mathcal{D}}}\Big[f(\omega(s,a))\Big]$$

$$(12)$$

Where $\omega(s,a)$ represents $\frac{d(s,a)}{d^{\mathcal{D}}(s,a)}$. Here we note that we exchange the summation of $s$ and $(s',a')$ in $(*)$ to get the final formulation. As is pointed out in (Sikchi et al., 2023), if we neglect the inner constraint $\omega \geq 0$, it's obvious that the above problem could be rewritten as:

$$\min_{V(s)} (1-\gamma)\mathbb{E}_{d_0(s)}[V(s)] + \mathbb{E}_{s,a\sim d^{\mathcal{D}}}\Big[f^*\big(r(s,a) + \gamma \sum_{s'} p(s'|s,a)V(s') - V(s)\big)\Big] \quad (13)$$

Here $f^*$ represents normal convex conjugate defined in eq (7). Although some previous methods directly use this formulation as their optimization target(Ma et al., 2022). The proper way to solve the inner constrained problem is reformulating it as:

$$\max_{\omega} \mathbb{E}_{s,a\sim d^{\mathcal{D}}}\Big[\omega(s,a)R(s,a,V)\Big] - \mathbb{E}_{s,a\sim d^{\mathcal{D}}}\Big[f(\omega(s,a))\Big] + \sum_{s,a} \mu(s,a)\omega(s,a) \quad (14)$$

Note that we discard the first term in eq (12) because it doesn't contain $\omega$, and we denote $R = r(s,a) + \gamma \sum_{s'} p(s'|s,a)V(s') - V(s)$. Since strong duality holds, we can directly solve KKT conditions and get the following 4 conditions: $\omega^*(s,a); \geq 0$; $\mu^*(s,a) \geq 0$; $f'(\omega^*(s,a)) = R(s,a,V) + \mu^*(s,a)$; $\mu^*(s,a)\omega^*(s,a) = 0$. Applying these conditions we can derive $\omega^*(s,a) = \max\{0, (f')^{-1}(R(s,a,V))\}$, which eliminates $\omega$ in eq (12) and gives the final optimization target:

V-DICE: $\min_{V(s)} (1-\gamma)\mathbb{E}_{d_0(s)}[V(s)] + \mathbb{E}_{s,a\sim d^{\mathcal{D}}}\Big[ \max\{0, (f')^{-1}(R)\} \cdot R - f(\max\{0, (f')^{-1}(R)\})\Big]$

$$(15)$$

Here $R$ is the abbreviation of $R(s,a,V)$. In this case, $f^*$ could be interpreted as $f^*(y) = \max\{0, (f')^{-1}(y)\} \cdot y - f\big(\max\{0, (f')^{-1}(y)\}\big)$, which is a variant of convex conjugate. Similarly, we can derive Q-DICE from eq (10) with Lagrangian duality(Nachum & Dai, 2020). It gives the following optimization target:

Q-DICE: $\max_{\pi} \min_{Q} \mathbb{E}_{d_0(s),\pi(a|s)}[Q(s,a)]$

$$+ \mathbb{E}_{s,a\sim d^{\mathcal{D}}}[f^*(r(s,a) + \gamma \sum_{s',a'} p(s'|s,a)\pi(a'|s')Q(s',a') - Q(s,a))] \quad (16)$$

Extracting policy is convenient for Q-DICE since its bi-level optimization procedure (Nachum et al., 2019b). For V-DICE, previous methods mainly adopt weighted-BC to extract $\pi^*$ (Lee et al., 2021; Sikchi et al., 2023), which maximizes the following objective with respect to $\pi$:

$$\mathbb{E}_{(s,a)\sim d^{\mathcal{D}}}[\omega^*(s,a)\log\pi(a|s)] \quad (17)$$

Besides that, Lee et al. (2021) proposed to use information projection for training policy, which minimizes the following objective:

$$\mathbb{KL}(d^{\mathcal{D}}(s)\pi(a|s)||d^{\mathcal{D}}(s)\pi^*(a|s)) = -\mathbb{E}_{\substack{s\sim d^{\mathcal{D}}, \\ a\sim\pi(\cdot|s)}}\left[\log\underbrace{\frac{d^*(s,a)}{d^{\mathcal{D}}(s,a)}}_{\omega^*(s,a)} - \log\frac{\pi(a|s)}{\pi_{\mathcal{D}}(a|s)} - \log\underbrace{\frac{d^*(s)}{d^{\mathcal{D}}(s)}}_{\text{constant for }\pi}\right]$$

As this method utilizes $a \sim \pi(\cdot|s)$, it could provide sufficient information even when $\pi$ deviates significantly from $\pi_{\mathcal{D}}$. However, it needs to estimate $\pi_{\mathcal{D}}$ with BC, which will introduce additional errors.

## B  PROOFS

**Theorem 1.** *In orthogonal-gradient update, $L_1^{\theta''}(s) - L_1^{\theta'}(s) = 0$ (under first order approximation).*

To give an intuitive explanation of this theorem, we can consider the directions of forward and backward gradients. If $\nabla_\theta L_2^\theta(s')$ happens to have component that is opposite to $\nabla_\theta L_1^\theta(s)$, then the effect of $\nabla_\theta L_1^\theta(s)$ will be offset by $\nabla_\theta L_2^\theta(s')$ because their component along the direction of $\nabla_\theta L_1^\theta(s)$ cancels out. Following the analysis before, we clearly don't want the effect of $\nabla_\theta L_1^\theta(s)$ to be weakened because it represents the process of finding optimal actions. It's worth noting that although $l_1^{\theta''}(s) - l_1^{\theta'}(s) < 0$ sometimes, we still don't want to preserve this effect of $\nabla_\theta L_2^\theta(s')$ because it may cause over-learn of minimizing $l_1^\theta(s)$.

*Proof.* For the sake of simplicity in our analysis, we will consider a first-order approximation when examining $l_1^{\theta''}(s) - l_1^{\theta'}(s)$. As adding $l_1^\theta(s) - l_1^\theta(s)$ will not change the target, we perform the following equivalent transformation:

$$
\begin{aligned}
l_1^{\theta''}(s) - l_1^{\theta'}(s) &= (l_1^{\theta''}(s) - l_1^\theta(s)) - (l_1^{\theta'}(s) - l_1^\theta(s)) \\
&= \nabla_\theta L_1^\theta(s)^\top (\theta'' - \theta) + \nabla_\theta L_1^\theta(s)^\top (\theta' - \theta) \\
&\quad + (\theta'' - \theta)^\top \frac{\nabla_\theta^2 l_1^{\xi''}(s)}{2} (\theta'' - \theta) + (\theta' - \theta)^\top \frac{\nabla_\theta^2 l_1^{\xi'}(s)}{2} (\theta' - \theta) \\
&= \nabla_\theta L_1^\theta(s)^\top (\theta'' - \theta') + O(\alpha^2)
\end{aligned}
\tag{18}
$$

Note that although two gradient steps are taken from $\theta$ to $\theta''$, their difference is still $O(\alpha)$. This property consequently renders all remainders as second-order infinitesimal.

Then by definition, we can substitute $\theta'' - \theta'$ with $-\alpha\eta \left( \nabla_\theta l_2^\theta(s') - \frac{\nabla_\theta L_1^\theta(s)^\top \nabla_\theta L_2^\theta(s')}{\|\nabla_\theta L_1^\theta(s)\|^2} \nabla_\theta L_1^\theta(s) \right)$ and get:

$$
l_1^{\theta''}(s) - l_1^{\theta'}(s) = -\alpha\eta \nabla_\theta L_1^\theta(s)^\top \left( \nabla_\theta l_2^\theta(s') - \frac{\nabla_\theta L_1^\theta(s)^\top \nabla_\theta L_2^\theta(s')}{\|\nabla_\theta L_1^\theta(s)\|^2} \nabla_\theta L_1^\theta(s) \right) = 0
\tag{19}
$$

$\square$

One can easily verify that when performing true-gradient, i.e. $\theta'' - \theta' = -\alpha \nabla_\theta L_2^\theta(s')$, we have:

$$
\begin{aligned}
l_1^{\theta''}(s) - l_1^{\theta'}(s) &\approx -\alpha \nabla_\theta L_1^\theta(s)^\top \nabla_\theta L_2^\theta(s') \\
&= \alpha (f_p^*)'(r + \gamma V(s') - V(s)) \cdot \nabla_\theta V(s) \cdot (f_p^*)'(r + \gamma V(s') - V(s)) \cdot \gamma \nabla_\theta V(s')^\top \\
&= \alpha\gamma \cdot [(f_p^*)'(r + \gamma V(s') - V(s))]^2 \nabla_\theta V(s)^\top \nabla_\theta V(s')
\end{aligned}
\tag{20}
$$

Due to the fact that $s$ and $s'$ represent consecutive states, it is likely that $\nabla_\theta V(s')$ and $\nabla_\theta V(s)$ fall within the same half-plane, resulting in $\nabla_\theta V(s)^\top \nabla_\theta V(s') > 0$. This interference results in an increase in $l_1^\theta(s)$ upon the application of the backward gradient, consequently leading to unlearning in the process of finding the best action. Gradient-orthogonalization prevents this unlearning by modifying backward gradient, which can preserve the information of backward gradient while eliminating interference.

**Definition 1**: We define $\nabla_\theta^\perp L_2^\theta(s')$ and $\nabla_\theta^\| L_2^\theta(s')$ as the perpendicular component and parallel component of $\nabla_\theta L_2^\theta(s')$ with respect to $\nabla_\theta L_1^\theta(s)$, formally:

$$\nabla_\theta^\perp L_2^\theta(s') = (f_p^*)'(r + \gamma V(s') - V(s)) \cdot \gamma \left( \underbrace{\nabla_\theta V(s') - \frac{\nabla_\theta V(s)^\top \nabla_\theta V(s')}{\|\nabla_\theta V(s)\|^2} \nabla_\theta V(s)}_{\nabla_\theta^\perp V(s')} \right)$$

$$\nabla_\theta^\| L_2^\theta(s') = (f_p^*)'(r + \gamma V(s') - V(s)) \cdot \gamma \underbrace{\frac{\nabla_\theta V(s)^\top \nabla_\theta V(s')}{\|\nabla_\theta V(s)\|^2} \nabla_\theta V(s)}_{\nabla_\theta^\| V(s')}$$

(21)

Similarly, we define $\nabla_\theta^\perp V(s')$ and $\nabla_\theta^\| V(s')$ as the perpendicular component and parallel component of $\nabla_\theta V(s')$ with respect to $\nabla_\theta V(s)$. Note that $\nabla_\theta^\perp V(s')$ and $\nabla_\theta^\| V(s')$ can directly obtained from $\nabla_\theta^\perp L_2^\theta(s')$ and $\nabla_\theta^\| L_2^\theta(s')$. So we directly define them in eq (21).

**Theorem 2.** *Define the angle between $\nabla_\theta V(s)$ and $\nabla_\theta V(s')$ as $\phi(s, s')$. In orthogonal-gradient update, if $\phi(s, s') \neq 0$, then for all $\eta > \frac{1}{\sin^2 \phi(s,s')} \left( \cos \phi(s, s') \frac{\|\nabla_\theta V(s)\|}{\gamma \|\nabla_\theta V(s')\|} - (\frac{\|\nabla_\theta V(s)\|}{\gamma \|\nabla_\theta V(s')\|})^2 \right)$, we have $\mathcal{L}^{\theta''}(s, s') - \mathcal{L}^\theta(s, s') < 0$.*

*Proof.* For simplicity of analysis, we still consider the change of $l^\theta(s, s')$ under first-order approximation. Due to the fact that the learning rate $\alpha$ is actually very small, we consider this approximation to be reasonable. Then we have:

$$l^{\theta''}(s, s') - l^\theta(s, s') = (\nabla_\theta L_1^\theta(s) + \nabla_\theta L_2^\theta(s'))^\top (\theta'' - \theta)$$
$$= -\alpha(\nabla_\theta L_1^\theta(s) + \nabla_\theta L_2^\theta(s'))^\top (\nabla_\theta L_1^\theta(s) + \eta \nabla_\theta^\perp L_2^\theta(s'))$$

(22)

First, consider a special case when $\nabla_\theta L_1^\theta(s) + \nabla_\theta^\| L_2^\theta(s')$ lies in the same direction as $\nabla_\theta L_1^\theta(s)$. In this case, $\left( \nabla_\theta L_1^\theta(s) + \nabla_\theta^\| L_2^\theta(s') \right)^\top \nabla_\theta L_1^\theta(s) \geq 0$. We can replace $\nabla_\theta L_2^\theta(s') = \nabla_\theta^\| L_2^\theta(s') + \nabla_\theta^\perp L_2^\theta(s')$ in eq (21) and get:

$$l^{\theta''}(s, s') - l^\theta(s, s') = -\alpha(\nabla_\theta L_1^\theta(s) + \nabla_\theta^\| L_2^\theta(s') + \nabla_\theta^\perp L_2^\theta(s'))^\top (\nabla_\theta L_1^\theta(s) + \eta \nabla_\theta^\perp L_2^\theta(s'))$$

$$= -\alpha \left( \underbrace{\left( \nabla_\theta L_1^\theta(s) + \nabla_\theta^\| L_2^\theta(s') \right)^\top \nabla_\theta L_1^\theta(s)}_{\text{in this case} \geq 0} + \eta \cdot \underbrace{\left( \nabla_\theta L_1^\theta(s) + \nabla_\theta^\| L_2^\theta(s') \right)^\top \nabla_\theta^\perp L_2^\theta(s')}_{=0} \right.$$

$$\left. + \underbrace{(\nabla_\theta^\perp L_2^\theta(s'))^\top \nabla_\theta L_1^\theta(s)}_{=0} + \underbrace{\eta \|\nabla_\theta^\perp L_2^\theta(s')\|^2}_{\text{by definition} \geq 0} \right)$$

(23)

It's obvious that in this case $l^{\theta''}(s, s') - l^\theta(s, s') \leq 0$ always satisfies, which means optimality is preserved as long as $\eta > 0$. This indicates that we only have to choose appropriate $\eta$ when $\left( \nabla_\theta L_1^\theta(s) + \nabla_\theta^\| L_2^\theta(s') \right)^\top \nabla_\theta L_1^\theta(s) = \|\nabla_\theta L_1^\theta(s)\|^2 + \nabla_\theta^\| L_2^\theta(s')^\top \nabla_\theta L_1^\theta(s) < 0$. Due to the fact that:

$$\nabla_\theta^\| L_2^\theta(s')^\top \nabla_\theta L_1^\theta(s) = -\gamma \cdot [(f_p^*)'(r + \gamma V(s') - V(s))]^2 \nabla_\theta^\| V(s')^\top \nabla_\theta V(s)$$
$$= -\gamma \cdot [(f_p^*)'(r + \gamma V(s') - V(s))]^2 \cos \phi \|\nabla_\theta V(s)\| \|\nabla_\theta V(s')\| < 0$$

(24)

We only have to consider the case when $\cos \phi > 0$. Note that the last equality can be derived by replacing $\nabla_\theta^\| V(s') = \cos \phi \|\nabla_\theta V(s')\| \frac{\nabla_\theta V(s)}{\|\nabla_\theta V(s)\|}$. In this case, we can simplify eq (22) with eq (21)

and the definition of $\nabla_\theta L_1^\theta(s)$ and $\nabla_\theta L_2^\theta(s')$. First plugging eq (21) into eq (22) we have:

$$
\begin{aligned}
l^{\theta''}(s, s') - l^\theta(s, s') = &-\alpha\Big(\nabla_\theta L_1^\theta(s) + \nabla_\theta L_2^\theta(s')\Big)^\top \Big[\nabla_\theta L_1^\theta(s) + \eta\Big(\nabla_\theta L_2^\theta(s')\\
&- \frac{\nabla_\theta L_1^\theta(s)^\top \nabla_\theta L_2^\theta(s')}{\|\nabla_\theta L_1^\theta(s)\|^2}\nabla_\theta L_1^\theta(s)\Big)\Big]\\
= &-\alpha(\|\nabla_\theta L_1^\theta(s)\|^2 + \nabla_\theta L_1^\theta(s)^\top \nabla_\theta L_2^\theta(s')\\
&+ \eta(\|\nabla_\theta L_2^\theta(s')\|^2 - (\frac{\nabla_\theta L_1^\theta(s)^\top \nabla_\theta L_2^\theta(s')}{\|\nabla_\theta L_1^\theta(s)\|})^2))
\end{aligned}
\tag{25}
$$

Recall that:

$$
\begin{aligned}
\nabla_\theta L_1^\theta(s) &= -(f_p^*)'(r + \gamma\overline{V}(s') - V(s))\nabla_\theta V(s)\\
\nabla_\theta L_2^\theta(s') &= \gamma(f_p^*)'(r + \gamma V(s') - \overline{V}(s))\nabla_\theta V(s')
\end{aligned}
\tag{26}
$$

We can plug eq (26) into eq (25) and get:

$$
\begin{aligned}
l^{\theta''}(s, s') - l^\theta(s, s') = &-\alpha\Big[(f_p^*)'(r + \gamma V(s') - V(s))\Big]^2\Big(\|\nabla_\theta V(s)\|^2 - \gamma\cdot\nabla_\theta V(s)^\top\nabla_\theta V(s')\\
&+ \eta\gamma^2(\|\nabla_\theta V(s')\|^2 - (\frac{\nabla_\theta V(s)^\top\nabla_\theta V(s')}{\|\nabla_\theta V(s)\|})^2)\Big)
\end{aligned}
\tag{27}
$$

To preserve optimality, we should choose appropriate $\eta$ to ensure this change is negative. We can get the equivalent form of this condition through some straightforward simplifications, which turns into:

$$
\begin{aligned}
\eta &\geq \frac{\gamma\cdot\nabla_\theta V(s)^\top\nabla_\theta V(s') - \|\nabla_\theta V(s)\|^2}{\gamma^2\cdot\|\nabla_\theta V(s')\|^2\cdot\sin^2\phi}\\
&= \frac{1}{\sin^2\phi}\left(\cos\phi\frac{\|\nabla_\theta V(s)\|}{\gamma\|\nabla_\theta V(s')\|} - (\frac{\|\nabla_\theta V(s)\|}{\gamma\|\nabla_\theta V(s')\|})^2\right)
\end{aligned}
\tag{28}
$$

$\square$

It is evident that this quadratic function has a global maximum when considering $\frac{\|\nabla_\theta V(s)\|}{\gamma\|\nabla_\theta V(s')\|}$ as a variable. In fact, the global maximum is exactly $\frac{\cot^2\phi}{4}$, so once $\phi(s, s') \neq 0$, there exists $\eta \geq \frac{\cot^2\phi}{4}$ such that $l^\theta(s, s')$ is still decrease after applying gradient-orthogonalization.

Although the result above gives a lower bound for $\eta$ independent of $\theta$, $s$, and $s'$, obviously the bound is far from tight. In fact, when we apply layer normalization (Ba et al., 2016) throughout optimization, $\|\nabla_\theta V(s)\|$ and $\|\nabla_\theta V(s')\|$ will be kept in a small interval. This means $\frac{\|\nabla_\theta V(s)\|}{\|\nabla_\theta V(s')\|}$ will be close to 1 and $\frac{\|\nabla_\theta V(s)\|}{\|\gamma\nabla_\theta V(s')\|}$ will be larger than 1. Under this condition, any positive choice of $\eta$ will satisfy the condition above and thus ensure a monotonic decrease of $L^\theta(s, s')$.

Finally, it can be easily deduced through similar reasoning that semi-gradient can't ensure the decrease of $l^\theta(s, s')$. So with gradient-orthogonalization, we can ensure the monotonic decrease of $L^\theta(s, s')$ while avoiding unlearning of minimizing $L_1^\theta(s)$.

**Definition 2**: We define $\phi_\theta(s, s')$ as the angle between $\nabla_\theta V(s)$ and $\nabla_\theta V(s')$, formally:

$$
\phi_\theta(s, s') = \arccos(\frac{\nabla_\theta V(s)^\top\nabla_\theta V(s')}{\|\nabla_\theta V(s)\|\cdot\|\nabla_\theta V(s')\|})
\tag{29}
$$

Note that with $\phi_\theta(s, s')$ we can represent $\nabla_\theta^\perp V(s')$ and $\nabla_\theta^\| V(s')$ more conveniently. One can easily verify $\nabla_\theta^\| V(s') = \cos\phi\|\nabla_\theta V(s')\|\frac{\nabla_\theta V(s)}{\|\nabla_\theta V(s)\|}$ and $\nabla_\theta^\perp V(s') = \sin\phi\|\nabla_\theta V(s')\|\frac{\nabla_\theta^\perp V(s')}{\|\nabla_\theta^\perp V(s')\|}$, this also corresponds to the geometric interpretation of $\nabla_\theta^\perp V(s')$ and $\nabla_\theta^\| V(s')$.

**Definition 3**: Define the Hessian of $V_\theta$ on $s$ and $s'$ as $H(s) = \nabla_\theta^2 V(s)$, $H(s') = \nabla_\theta^2 V(s')$. Then define $\Delta\lambda(A) = \lambda_{\min}(A) - \lambda_{\max}(A)$, where $\lambda_{\min}(A)$ and $\lambda_{\max}(A)$ are minimal and maximal

eigenvalues of $A$ respectively. Finally define the constant $\beta$ appears in Theorem 3 with $\beta = \frac{3\sqrt{3}}{4} \cdot \min\left\{\Delta\lambda\Big(H(s)\Big), \Delta\lambda\Big(H(s')\Big)\right\}$. Note that when $V_\theta$ is smooth with respect to $\theta$, $\beta$ is close to 0.

**Lemma 1**: When considering inner constraint $\omega \geq 0$ in eq (12), i.e. $f^*(y) = \max\{0, (f')^{-1}(y)\} \cdot y - f\big(\max\{0, (f')^{-1}(y)\}\big)$. The inequality $(f^*)'(y) \geq 0$ holds for arbitrary convex $f$ and $y$.

*Proof.* First considering the case when $(f')^{-1}(y) \leq 0$, it's obvious that $f^*(y) = -f(0)$. In this case $(f^*)'(y) = 0$, which satisfy the inequality.

Then considering $(f')^{-1}(y) > 0$, in this case $f^*(y) = (f')^{-1}(y) \cdot y - f\big((f')^{-1}(y)\big)$. Note that $y \cdot x - f(x)$ is concave, which means its saddle point $x = (f')^{-1}(y)$ is also the global maximum point. Plugging $x = (f')^{-1}(y)$ into $y \cdot x - f(x)$ leads to the equation:

$$\sup\{y \cdot x - f(x), x \in \mathbb{R}\} = (f')^{-1}(y) \cdot y + f\big((f')^{-1}(y)\big) = f^*(y) \tag{30}$$

This means in this case $f^*$ is exactly the normal convex conjugate, while its solution $(f')^{-1}(y)$ is ensured to be positive. Then by definition, we have:

$$
\begin{aligned}
(f^*)'(y) &= \frac{(f^*)(y + \Delta y) - (f^*)(y)}{\Delta y} \\
&= \frac{\Big(\max\limits_x (y + \Delta y) \cdot x - f(x)\Big) - \Big((f')^{-1}(y) \cdot y - f((f')^{-1}(y))\Big)}{\Delta y} \\
&\geq \frac{\Big((f')^{-1}(y) \cdot (y + \Delta y) - f((f')^{-1}(y))\Big) - \Big((f')^{-1}(y) \cdot y - f((f')^{-1}(y))\Big)}{\Delta y} \\
&= (f')^{-1}(y) > 0
\end{aligned}
\tag{31}
$$

So for arbitrary convex $f$ and $y$, we have $(f^*)'(y) \geq 0$. $\qquad\square$

**Lemma 2**: If $x$ and $y$ are $n$-dimensional orthonomal vectors, $A$ is a $n \times n$ real symmetric matrix, then $x^\top A y \geq \lambda_{\min} - \lambda_{\max}$.

*Proof.* As $A$ is a real symmetric matrix, it can establish a set of orthonormal basis vectors using its eigenvectors. Suppose these vectors are $z_{i=1\cdots n}$. We can decompose $x$ and $y$ under these basis vectors, $x = \sum_{i=1}^n a_i z_i$ and $y = \sum_{j=1}^n b_j z_j$ respectively. Then we have:

$$\sum_{i=1}^n a_i b_i = \underbrace{x^\top y = 0}_{\text{orthonomal vectors}} \tag{32}$$

$$x^\top A y = x^\top (\sum_{j=1}^n \lambda_j b_j z_j) = (\sum_{i=1}^n a_i z_i)(\sum_{j=1}^n \lambda_j b_j z_j) = \sum_{i=1}^n \lambda_i a_i b_i \tag{33}$$

Which means $x^\top A y$ just reweight $a_i b_i$ with $\lambda_i$ and sum them together. It's clear that:

$$\sum_{i=1}^n \lambda_i a_i b_i \geq \lambda_{\max}(\sum_{j \text{ s.t. } a_j b_j < 0} a_j b_j) + \lambda_{\min}(\sum_{j \text{ s.t. } a_j b_j > 0} a_j b_j) \tag{34}$$

So with $\sum_{j \text{ s.t. } a_j b_j < 0} a_j b_j + \sum_{j \text{ s.t. } a_j b_j > 0} a_j b_j = 0$ we have

$$\sum_{i=1}^n \lambda_i a_i b_i \geq (\lambda_{\min} - \lambda_{\max}) \sum_{j \text{ s.t. } a_j b_j > 0} a_j b_j \geq (\lambda_{\min} - \lambda_{\max})\|x\|\|y\| = (\lambda_{\min} - \lambda_{\max}) \tag{35}$$

$\qquad\square$

**Theorem 3** (Orthogonal-gradient update helps alleviate feature co-adaptation). *Assume the norm of $\nabla_\theta V(s)$ is bounded, i.e. $\forall s, m \leq \|\nabla_\theta V(s)\|^2 \leq M$. Define consecutive value curvature $\xi(\theta, s, s') = \nabla_\theta^\perp V(s')^\top H(s) \nabla_\theta^\perp V(s')$, where $H(s)$ is the Hessian matrix of $V(s)$. Assume $\xi(\theta, s, s') \geq l \cdot \|\nabla_\theta^\perp V(s')\|^2$ where $l > 0$. Then in orthogonal-gradient update, we have*

$$\Psi_{\theta''}(s, s') - \Psi_{\theta'}(s, s') \leq -\alpha\eta\gamma \cdot (f^*)'(r + \gamma V(s') - V(s))[\sin^2 \phi(s, s') \cdot l \cdot m + \beta \cdot M] \quad (36)$$

*where $\beta$ is a constant close to 0 if the condition number of $H(s)$ is small.*

*Proof.* To check the change of $\psi_\theta(s, s')$ caused by backward gradient, we also conduct analysis under first-order approximation. This is reasonable due to the fact that $\theta'' - \theta$ is on the order of $O(\alpha)$, where $\alpha$ typically represents a very small learning rate. So applying the same transformation in Theorem 1 we can get:

$$\begin{aligned}
\Psi_{\theta''}(s, s') - \Psi_{\theta'}(s, s') &= \nabla_\theta \Psi_{\theta'}(s, s')^\top (\theta'' - \theta') + O(\alpha^2) \\
&= \left[\nabla_\theta V(s')^\top H(s) + \nabla_\theta V(s)^\top H(s')\right](\theta'' - \theta') + O(\alpha^2)
\end{aligned} \quad (37)$$

Where $H(s)$ is the Hessian matrix of $V_\theta(s)$ with respect to $\theta$, i.e. $\nabla_\theta^2 V_\theta(s)$. Note that we substitute $\nabla_\theta \Psi_{\theta'}(s, s')$ with $\nabla_\theta \Psi_\theta(s, s')$ because their difference will only contribute to a second-order infinitesimal with respect to $\alpha$, which will be absorbed into $O(\alpha^2)$.

After replacing $\theta'' - \theta'$ with $-\alpha\eta\nabla_\theta l^\perp(s')$ we have

$$\begin{aligned}
\Psi_{\theta''}(s, s') - \Psi_{\theta'}(s, s') &\approx -\alpha\eta\gamma \cdot (f_p^*)'(r + \gamma V(s') - V(s))\Big(\nabla_\theta V(s')^\top H(s) \nabla_\theta^\perp V(s') \\
&\quad + \nabla_\theta V(s)^\top H(s') \nabla_\theta^\perp V(s')\Big) \\
&= -\alpha\eta\gamma \cdot (f_p^*)'(r + \gamma V(s') - V(s))\Big(\underbrace{\|\nabla_\theta^\perp V(s')\|_{H(s)}^2}_{①} \\
&\quad + \underbrace{\langle \nabla_\theta^\| V(s'), \nabla_\theta^\perp V(s') \rangle_{H(s)}}_{②} + \underbrace{\langle \nabla_\theta V(s), \nabla_\theta^\perp V(s') \rangle_{H(s')}}_{③}\Big)
\end{aligned} \quad (38)$$

One can further utilize the following equation to replace $\nabla_\theta^\| V(s')$ and $\nabla_\theta^\perp V(s')$ in ②, ③:

$$\nabla_\theta^\| V(s') = |\cos\phi|\|\nabla_\theta V(s')\|\frac{\cos\phi \nabla_\theta V(s)}{|\cos\phi|\|\nabla_\theta V(s)\|}, \quad \nabla_\theta^\perp V(s') = \sin\phi\|\nabla_\theta V(s')\|\frac{\nabla_\theta^\perp V(s')}{\|\nabla_\theta^\perp V(s')\|} \quad (39)$$

This leads to the following results:

$$\begin{aligned}
② &= \sin\phi|\cos\phi|\|\nabla_\theta V(s')\|^2 \langle \frac{\nabla_\theta^\perp V(s')}{\|\nabla_\theta^\perp V(s')\|}, \frac{\cos\phi \nabla_\theta V(s)}{|\cos\phi|\|\nabla_\theta V(s)\|} \rangle_{H(s)} \\
③ &= \sin\phi\|\nabla_\theta V(s)\|\|\nabla_\theta V(s')\| \langle \frac{\nabla_\theta^\perp V(s')}{\|\nabla_\theta^\perp V(s')\|}, \frac{\nabla_\theta V(s)}{\|\nabla_\theta V(s)\|} \rangle_{H(s')}
\end{aligned} \quad (40)$$

Note that as $\frac{\cos\phi \nabla_\theta V(s)}{|\cos\phi|\|\nabla_\theta V(s)\|}, \frac{\nabla_\theta V(s)}{\|\nabla_\theta V(s)\|}$ are orthonormal to $\frac{\nabla_\theta^\perp V(s')}{\|\nabla_\theta^\perp V(s')\|}$ and $H(s)$ is real symmetric, ② and ③ can be futher bounded with lemma 2. This gives the following bound:

$$\begin{aligned}
② + ③ &\geq \sin\phi|\cos\phi|\|\nabla_\theta V(s')\|^2(\lambda_{\min}(H(s)) - \lambda_{\max}(H(s))) \\
&\quad + \sin\phi\|\nabla_\theta V(s)\|\|\nabla_\theta V(s')\|(\lambda_{\min}(H(s')) - \lambda_{\max}(H(s'))) \\
&\geq (\sin\phi|\cos\phi|\|\nabla_\theta V(s')\|^2 + \sin\phi\|\nabla_\theta V(s)\|\|\nabla_\theta V(s')\|) \cdot \min\Big\{\Delta\lambda\Big(H(s)\Big), \Delta\lambda\Big(H(s')\Big)\Big\}
\end{aligned} \quad (41)$$

The $\Delta\lambda$ here is defined in Definition 3. With $\forall s, m \leq \|\nabla_\theta V(s)\|^2 \leq M$ we can further bound ② + ③ with:

$$② + ③ \geq (1 + |\cos\phi|)\sin\phi \cdot \min\Big\{\Delta\lambda\Big(H(s)\Big), \Delta\lambda\Big(H(s')\Big)\Big\} \cdot M \quad (42)$$

Moreover, due to the fact that $\phi \in [0, \pi]$, $\sin \phi(1 + |\cos \phi|)$ is non-negative and have upper bound $\frac{3\sqrt{3}}{4}$.

Under the assumption of consecutive value curvature, and by replacing ② + ③ in eq (38) with eq (42), $\Psi_{\theta''}(s, s') - \Psi_{\theta'}(s, s')$ can be ultimately bounded by:

$$\Psi_{\theta''}(s, s') - \Psi_{\theta'}(s, s') \leq -\alpha\eta\gamma \cdot (f_p^*)'(r + \gamma V(s') - V(s))[\sin^2 \phi \cdot l \cdot m + \beta \cdot M] \qquad (43)$$

$$\square$$

It's worth noting that we can derive the final bound because $(f_p^*)'(r + \gamma V_\theta(s') - V_\theta(s))$ is always non-negative, which has been proved in Lemma 1. When minimizing standard mean-square Bellman error, $r + \gamma V_\theta(s') - V_\theta(s)$ could be negative and thus no upper bound is guaranteed. This means Theorem 3 reflects the nature of the combination of DICE and orthogonal-gradient, rather than the nature of each one separately.

**Definition 4**: Define $\mathcal{K}_{\max}$ as the maximum value of the square of $\nabla_s \nabla_\theta V_\theta(s)$'s eigenvalues, i.e. $\mathcal{K}_{\max} = \max\left[\lambda\big(\nabla_s \nabla_\theta V_\theta(s)\big)^2\right]$. Here we inherit $\lambda(\cdot)$ from Definition 3. Similarly define $\mathcal{K}_{\min}$. Then define the constant $C$ appears in Proposition 2.1 with $C = \frac{\mathcal{K}_{\min}^2}{2\mathcal{K}_{\max}}$. Note that $C \geq 0$ and $C$ is almost surely positive unless 0 is exactly $\nabla_s \nabla_\theta V_\theta(s)$'s eigenvalue.

**Theorem 4.** (How feature co-adaptation affects state-level robustness) *Under linear setting when $V(s)$ can be expressed as $\nabla_\theta V(s)^\top \theta$, assume $\|\nabla_\theta V(s)\|^2 \leq M$, then there exists some small noise $\varepsilon \in \mathbb{R}^{|S|}$ such that $V(s' + \varepsilon) - V(s)$ will have different sign with $V(s') - V(s)$, if $\Psi_\theta(s, s') > M - C \cdot \|\varepsilon\|^2$ for some constant $C > 0$.*

*Proof.* We begin with deriving the upper bound for $\|\nabla_\theta V(s) - \nabla_\theta V(s')\|^2$. Expanding $\|\nabla_\theta V(s) - \nabla_\theta V(s')\|^2$ we can get:

$$\|\nabla_\theta V(s) - \nabla_\theta V(s')\|^2 = \|\nabla_\theta V(s)\|^2 + \|\nabla_\theta V(s')\|^2 - 2 \cdot \nabla_\theta V(s')^\top \nabla_\theta V(s) \qquad (44)$$

Plugging in the bound $\|\nabla_\theta V(s)\|^2 \leq M$ and $\nabla_\theta V(s')^\top \nabla_\theta V(s) > M - \frac{\mathcal{K}_{\min}^2}{2\mathcal{K}_{\max}}\|\varepsilon\|^2$ we have

$$\|\nabla_\theta V(s) - \nabla_\theta V(s')\|^2 < \frac{\mathcal{K}_{\min}^2}{\mathcal{K}_{\max}}\|\varepsilon\|^2 \qquad (45)$$

Without loss of generality, we only analyze the case when $V(s') - V(s) < 0$. Due to the fact that noise $\varepsilon$ is usually small, we can use first-order Taylor expansion to approximate $V(s' + \varepsilon)$ in the neighborhood of $V(s')$. Then we have:

$$\begin{aligned} V(s' + \varepsilon) &= \nabla_\theta V(s' + \varepsilon)^\top \theta \\ &\approx \left[\nabla_\theta V(s') + \nabla_s \nabla_\theta V_\theta(s)^\top \varepsilon\right]^\top \theta \\ &= \nabla_\theta V(s')^\top \theta + \varepsilon^\top A\theta \end{aligned} \qquad (46)$$

Here we denote $A$ as $\nabla_s \nabla_\theta V_\theta(s)$ for simplicity. After substituting eq (46) into $V(s' + \varepsilon) - V(s)$ we have:

$$\begin{aligned} V_\theta(s' + \varepsilon) - V_\theta(s) &= \nabla_\theta V(s')^\top \theta + \varepsilon^\top A\theta - \nabla_\theta V(s)^\top \theta \\ &= \left[\nabla_\theta V(s') - \nabla_\theta V(s)\right]^\top \theta + \varepsilon^\top A\theta \\ &= \Delta f^\top \theta + \varepsilon^\top A\theta \end{aligned} \qquad (47)$$

Here again, for simplicity, we denote the difference of feature $\nabla_\theta V(s') - \nabla_\theta V(s)$ as $\Delta f$. Similarly we can get $V(s') - V(s) = \Delta f^\top \theta$. Then set $\varepsilon = -\|\varepsilon\|\frac{(AA^\top)^{-1}A\Delta f}{\|(AA^\top)^{-1}A\Delta f\|}$, which could be interpreted as a rotated $\varepsilon$. We'll then prove that this rotated $\varepsilon$ can change the sign of $V(s' + \varepsilon) - V(s)$ compared to $V(s') - V(s)$ while still satisfying $\Psi_\theta(s, s') > M - \frac{\mathcal{K}_{\min}^2}{2\mathcal{K}_{\max}} \cdot \|\varepsilon\|^2$.

Consider the following optimization problem:

$$\min_x \|A^\top x - \Delta f\|^2 \qquad (48)$$

Solving this problem we can get the $x$ whose projection under $A$ is closest to $\Delta f$. It's well known that this problem has the following close-form solution:

$$x = (AA^\top)^{-1} A \cdot \Delta f \tag{49}$$

Then under the condition that $\Delta f \in \text{span}(A)$, it's obvious that we can choose $x$ such that this optimization target is 0. This also means that $A^\top (AA^\top)^{-1} A \cdot \Delta f = \Delta f$. Plugging in $\varepsilon = -\|\varepsilon\| \frac{(AA^\top)^{-1} A \Delta f}{\|(AA^\top)^{-1} A \Delta f\|}$ and the above relationship into eq (47) we have:

$$V(s' + \varepsilon) - V(s) = \Delta f^\top \theta - \|\varepsilon\| \cdot \Delta f^\top \left( \frac{A^\top (AA^\top)^{-1} A}{\|(AA^\top)^{-1} A \Delta f\|} \right) \theta$$

$$= (1 - \frac{\|\varepsilon\|}{\|(AA^\top)^{-1} A \Delta f\|}) \cdot \Delta f^\top \theta \tag{50}$$

Recall that from eq (45) we have $\|\Delta f\| = \|\nabla_\theta V(s) - \nabla_\theta V(s')\| < \frac{\mathcal{K}_{\min}}{\sqrt{\mathcal{K}_{\max}}} \|\varepsilon\|$. Applying this to eq (50) we have:

$$\frac{\|\varepsilon\|}{\|(AA^\top)^{-1} A \Delta f\|} > \frac{\|\varepsilon\|}{\max \left| \lambda \left[ (AA^\top)^{-1} A \right] \right| \cdot \|\Delta f\|}$$

$$> \frac{\sqrt{\mathcal{K}_{\max}}}{\mathcal{K}_{\min} \cdot \max \left| \lambda \left[ (AA^\top)^{-1} A \right] \right|}$$

$$= \sqrt{\frac{\mathcal{K}_{\max}}{\mathcal{K}_{\min}^2} \frac{1}{\max \lambda \left( A^\top \left[ (AA^\top)^{-1} \right]^2 A \right)}} \tag{51}$$

From the definition of $\mathcal{K}_{\max}$ and $\mathcal{K}_{\min}$ it's obvious that $\lambda \left( A^\top \left[ (AA^\top)^{-1} \right]^2 A \right) < \frac{\mathcal{K}_{\max}}{\mathcal{K}_{\min}^2}$. Plugging this into eq (51) we have $\frac{\|\varepsilon\|}{\|(AA^\top)^{-1} A \Delta f\|} > 1$, which means:

$$V(s' + \varepsilon) - V(s) = (1 - \frac{\|\varepsilon\|}{\|(AA^\top)^{-1} A \Delta f\|}) \cdot \Delta f^\top \theta > 0 \tag{52}$$

The last inequality comes from the case we consider that $V(s') - V(s) = \Delta f^\top \theta < 0$. Moreover, due to the fact that the choice of $\varepsilon$ keeps its norm unchanged, $\Psi_\theta(s, s') > M - \frac{\mathcal{K}_{\min}^2}{2\mathcal{K}_{\max}} \cdot \|\varepsilon\|^2$ still holds. Finally, one can easily verify this under the case where $V(s') - V(s) > 0$. $\square$

## C  MORE DISCUSSION ON GRADIENT FLOW OF DICE

In this section, we'll delve into discussing the first gradient term mentioned in section 2.2. We'll first verify that adding $(1-\gamma)\mathbb{E}_{s \sim d_0}[\nabla_\theta V(s)]$ won't change our theoretical results. Then we'll show that in our practical algorithm, this term will not impact the results of projection, and thus also has no influence on the practical algorithm of O-DICE.

To begin with, we first consider Theorem 1, Theorem 3 and Theorem 4. Theorem 1 shows that orthogonal-gradient will not influence the process of seeking the best actions while Theorem 3, 4 shows orthogonal-gradient leads to better state feature representation and thus brings more robustness. These two theorems are completely independent of the first gradient term. As for Theorem 2, it shows that $\nabla_\theta L_1^\theta(s) + \nabla_\theta^\perp L_2^\theta(s')$ lies in the same half-plane with $\nabla_\theta L_1^\theta(s) + \nabla_\theta L_2^\theta(s')$. One can easily verify that after adding the first term, the extended orthogonal-gradient $(1 - \gamma)\mathbb{E}_{s \sim d_0}[\nabla_\theta V(s)] + \nabla_\theta L_1^\theta(s) + \nabla_\theta^\perp L_2^\theta(s')$ still lies in the same half-plane with the extended true-gradient $(1 - \gamma)\mathbb{E}_{s \sim d_0}[\nabla_\theta V(s)] + \nabla_\theta L_1^\theta(s) + \nabla_\theta L_2^\theta(s')$. This indicates that the extended version of Theorem 2 also holds even considering $(1 - \gamma)\mathbb{E}_{s \sim d_0}[\nabla_\theta V(s)]$.

Moving forward to the practical algorithm, because we replace $d_0$ with $d^\mathcal{D}$ and use empirical Bellman operator, only one batch of $(s, a, s')$ is needed for one gradient step. This makes $(1-\lambda)\nabla_\theta V(s)$ has exactly same direction as $\nabla_\theta L_1^\theta(s)$. Due to the property of orthogonal projection, $\nabla_\theta^\perp L_2^\theta(s')$ keeps the same after considering the first gradient term. Thus it has no influence both theoretically and practically.

# D    EXPERIMENTAL DETAILS

**More details of the practical algorithm**    As is shown in the pseudo-code, O-DICE doesn't need parameters for policy extraction , which is more convenient when compared with many other state-of-the-art algorithms (Xu et al., 2023; Garg et al., 2023). Moreover, as we choose Pearson $\chi^2$ for $f$-divergence in practice, the corresponding $f$, $f^*$, $(f')^{-1}$ has the following form:

$$f(x) = (x - 1)^2; \; f^*(y) = y(\frac{y}{4} + 1); \; (f')^{-1}(R) = \frac{R}{2} + 1 \tag{53}$$

We also apply the trick used in SQL (Xu et al., 2023), which removes the "+1" term in $\omega^*(s, a)$ in eq (17). This trick could be seen as multiplying the residual term $R$ by a large value. Besides that, during policy extraction, we use $r(s, a) + \gamma V_{\bar{\theta}}(s') - V_\theta(s)$ to represent residual term due to its relationship with seeking optimal actions. The final objective for policy extraction in our method is $\mathbb{E}_{(s,a)\sim d^{\mathcal{D}}}[\max\{0, r(s, a) + \gamma V_{\bar{\theta}}(s') - V_\theta(s)\} \cdot \log \pi(a|s)]$.

**Toy case experimental details**    In order to study the influence of forward and backward gradient on $V(s)$, we ran V-DICE with 3 different gradient types (true-gradient, semi-gradient, orthogonal-gradient) separately on a pre-collected dataset for 10000 steps and plotted their corresponding $V(s)$ distributions. The pre-collected dataset contains about 20 trajectories, 800 transitions. For the network, we use 3-layer MLP with 128 hidden units, Adam optimizer (Kingma & Ba, 2015) with a learning rate of $10^{-4}$. We choose $5 \cdot 10^{-3}$ as the soft update weight for $V$. Moreover, as mentioned before, V-DICE with different gradient types may need different $\lambda$ and $\eta$ to achieve the best performance. For the sake of fairness, we tuned hyper parameters for all gradient types to achieve the best performance.

**D4RL experimental details**    For all tasks, we conducted our algorithm for $10^6$ steps and reported the final performance. In MuJoCo locomotion tasks, we computed the average mean returns over 10 evaluations every $5 \cdot 10^3$ training steps, across 5 different seeds. For AntMaze tasks, we calculated the average over 50 evaluations every $2 \cdot 10^4$ training steps, also across 5 seeds. Following previous research, we standardized the returns by dividing the difference in returns between the best and worst trajectories in MuJoCo tasks. In AntMaze tasks, we subtracted 3 from the rewards.

For the network, we use 3-layer MLP with 256 hidden units and Adam optimizer (Kingma & Ba, 2015) with a learning rate of $1 \cdot 10^{-4}$ for both $V$ and $\pi$ in all tasks. We use a target network with soft update weight $5 \cdot 10^{-3}$ for $V$.

We re-implemented OptiDICE (Lee et al., 2021) using PyTorch and ran it on all datasets. We followed the same reporting methods as mentioned earlier. Although $f$-DVL (Sikchi et al., 2023) reported the performance of two variants, we only take the results of $f$-DVL using $\chi^2$ because it chooses similar $f$ as other DICE methods. Baseline results for other methods were directly sourced from their respective papers.

In O-DCIE, we have two parameters: $\lambda$ and $\eta$. Because a larger $\lambda$ indicates a stronger ability to search for optimal actions, we select a larger $\lambda$ if the value of $V$ doesn't diverge. As for $\eta$, we experiment with values from the range $[0.2, 0.4, 0.6, 0.8, 1.0]$ to find the setting that gives us the best performance. The values of $\lambda$ and $\eta$ for all datasets are listed in Table 3. In offline IL, we use $\lambda = 0.4$ and $\eta = 1.0$ across all datasets.

Table 3: $\lambda$ and $\eta$ used in O-DICE

| Dataset | $\lambda$ | $\eta$ |
|---|---|---|
| halfcheetah-medium-v2 | 0.5 | 0.2 |
| hopper-medium-v2 | 0.6 | 1.0 |
| walker2d-medium-v2 | 0.5 | 0.2 |
| halfcheetah-medium-replay-v2 | 0.6 | 0.2 |
| hopper-medium-replay-v2 | 0.6 | 1.0 |
| walker2d-medium-replay-v2 | 0.6 | 0.6 |
| halfcheetah-medium-expert-v2 | 0.5 | 0.2 |
| hopper-medium-expert-v2 | 0.5 | 1.0 |
| walker2d-medium-expert-v2 | 0.5 | 0.2 |
| antmaze-umaze-v2 | 0.6 | 1.0 |
| antmaze-umaze-diverse-v2 | 0.4 | 1.0 |
| antmaze-medium-play-v2 | 0.7 | 0.2 |
| antmaze-medium-diverse-v2 | 0.7 | 0.2 |
| antmaze-large-play-v2 | 0.7 | 0.8 |
| antmaze-large-diverse-v2 | 0.8 | 0.8 |

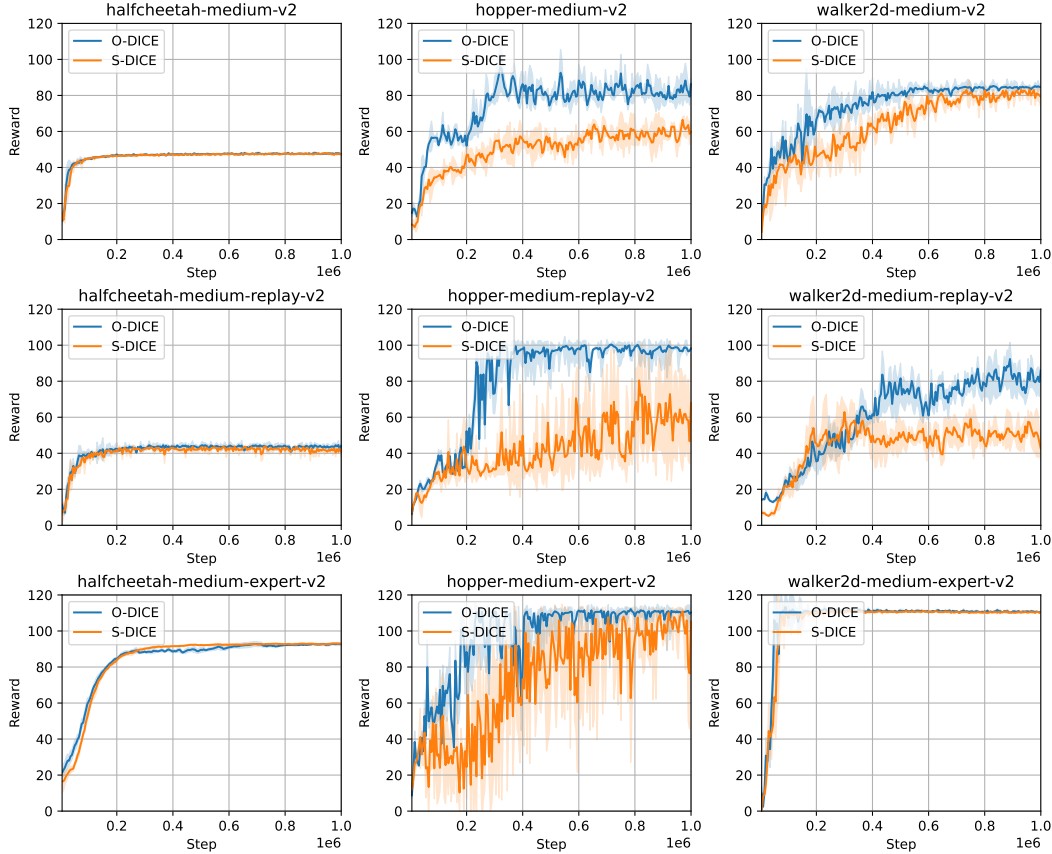

Figure 4: Learning curves of O-DICE and S-DICE on D4RL MuJoCo locomotion datasets.

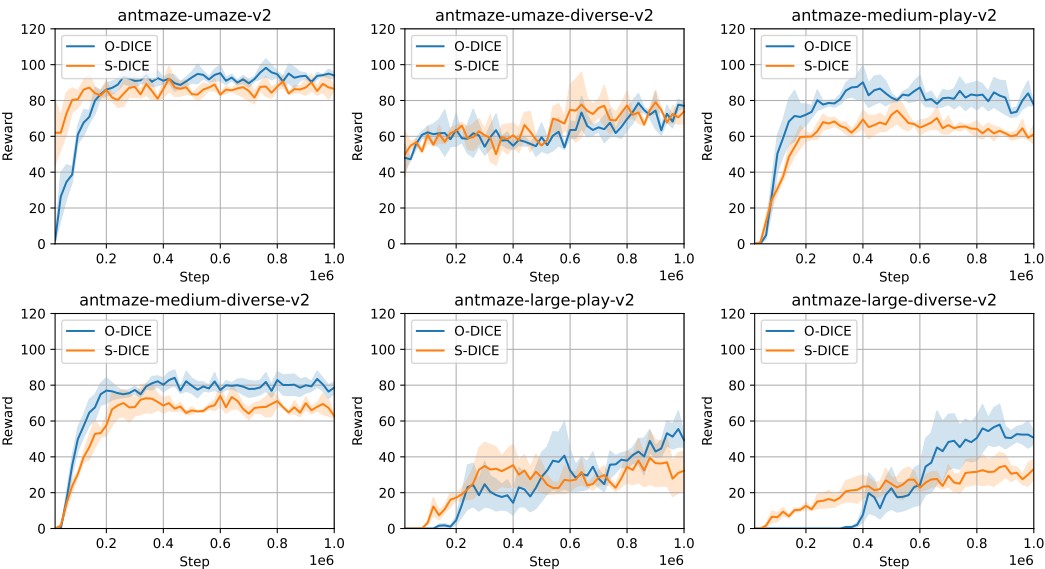

Figure 5: Learning curves of O-DICE and S-DICE on D4RL AntMaze datasets.

