# OpenReview forum: "ODICE: Revealing the Mystery of Distribution Correction Estimation via Orthogonal-gradient Update"
_ICLR.cc/2024/Conference — ICLR 2024 spotlight_

### Official Review · Reviewer_g7U6 · 2023-10-30

**Soundness:** 3 good
**Presentation:** 3 good
**Contribution:** 3 good
**Rating:** 8
**Confidence:** 4

**Summary:**

This paper uncovers an explanation for a mysterious phenomenon among off-policy RL DICE methods: state-action level behavior constraints are more principled but action-level behavior constraints perform better empirically. The authors show how an action-level behavior constraint can be achieved from two different view points: Exponential Q-Learning (semi-gradient) designed to specifically impose an action-level constraint and the forward-gradient view put forth by the authors. This then implies that the key difference between the semi-gradient and true-gradient updates (true-DICE) is the backward-gradient. Given both the superior empirical performance of the forward gradient approaches and the desire to better match the more principled state-action approach, the authors propose to retain the backward-gradient component albeit with a twist: include the backward component but reject any part of it which aligns with the forward component (thereby removing any gradient cancellation). The authors then support this technique both theoretically (alleviates interference of forward/backward updates and feature co-adaption) and with extensive empirical validation (Fig 2 shows the method avoids overestimating V(s) outside the dataset distribution, Table 1 and Table 2 show it performs well on difficult tasks, and Fig 3 shows its worst-case performance is higher than other approaches).

**Strengths:**

I thought this paper did an excellent job of identifying and isolating a key problem, making an astute connection to research elsewhere in the literature (EQL), and then exploiting this connection to develop and evaluate a simple fix. I think the paper is mostly well-written with good theoretical and empirical support. I have a few questions regarding one of the theoretical statements and one of the figures, but otherwise have no other major concerns.

**Weaknesses:**

I would like to see a bit more explanation / background in the second-to-last paragraph of section 1 (Introduction) before you start discussing the bellman residual term and *true-gradient* methods.

**Questions:**

- In section 3, first paragraph, did you mean "orthogonal-gradient" instead of "vertical-gradient"?
- Theorem 3: Given that you measure feature co-adaption with an un-normalized dot-product, couldn't the result in Theorem 3 be achieved by a shrinking of the feature vectors rather than a change in their orientation or relative representations? I assume your intention is to show that $\nabla_{\theta} V(s)$ and $\nabla_{\theta} V(s')$ are more different (or separated) under the orthogonal-gradient update (i.e., $\theta=\theta''$) than they are normally (i.e., $\theta=\theta'$).
- Figure 2: I can see how (d) aligns well with the dataset support, but so does (b). Is (b) undesireable because the actual values V(s) across the offline states are poorly approximated (i.e., the gradient across V(s) is uninformative)? If so, can you please add that to the caption?
- You say "Hence, Eq.(36) < 0" above Theorem 4? Is this a typo? Did you mean to refer to equation 6?

---

> ### Author Response · Authors · 2023-11-18
>
> We thank the reviewer for the effort engaged in the review phase and the constructive comments. Regarding the questions, we provide the detailed responses separately as follows.
>
> >Questions:
> In section 3, first paragraph, did you mean "orthogonal-gradient" instead of "vertical-gradient"?
>
> We are sorry that we mistakenly wrote "orthogonal-gradient" as "vertical-gradient". We will fix this typo in the updated version.
>
> >Theorem 3: Given that you measure feature co-adaption with an un-normalized dot-product, couldn't the result in Theorem 3 be achieved by a shrinking of the feature vectors rather than a change in their orientation or relative representations? I assume your intention is to show that  and  are more different (or separated) under the orthogonal-gradient update (i.e., ) than they are normally (i.e., ).
>
> We acknowledge that shrinking the feature vectors can, to some extent, decrease $\Psi_\theta(s, s')$. However, without introducing an auxiliary optimization objective to DICE methods, $\| \nabla_\theta V(s) \|$ is often bounded, as assumed in Theorem 3 with $m<| \nabla_\theta V(s) |<M$. In fact, when employing practical tricks for training stability, such as layer normalization[1], $\| \nabla_\theta V(s) \|$ is even confined to a very small interval. Therefore, a smaller $\Psi_\theta(s, s')$ typically indicates that feature vectors are more orthogonal in orientation rather than simply having a smaller norm.
>
> [1]Layer normalization. Lei Jimmy Ba, Jamie Ryan Kiros, and Geoffrey E. Hinton.
>
> >Figure 2: I can see how (d) aligns well with the dataset support, but so does (b). Is (b) undesireable because the actual values V(s) across the offline states are poorly approximated (i.e., the gradient across V(s) is uninformative)? If so, can you please add that to the caption?
>
> We apologize for not including this in the caption. You are correct, and we will add it in the updated version.
>
> >You say "Hence, Eq.(36) < 0" above Theorem 4? Is this a typo? Did you mean to refer to equation 6?
>
> We are sorry that we mistakenly wrote "Eq.(6)" as "Eq.(36)" here and it will be addressed in the updated version.

---

> > ### Comment · Reviewer_g7U6 · 2023-11-20
> > **Follow-Up**
> >
> > Thank you for your explanations. I maintain my score.
> >
> > Re shrinking vs rotating feature vectors, I would suggest adding this nuanced explanation to the appendix and referring the reader there from the main body via a brief comment acknowledging that e.g., "decrease in feature vector norm is unlikely (see Appx)".

---

> ### Author Response · Authors · 2023-11-23
>
> Thanks for your suggestions. We will add it in the updated version.

---

### Official Review · Reviewer_SNRk · 2023-11-01

**Soundness:** 3 good
**Presentation:** 3 good
**Contribution:** 3 good
**Rating:** 6
**Confidence:** 4

**Summary:**

- The paper identifies two gradient terms when learning the value function using true gradients: the forward gradient (taken on the current state) and the backward gradient (taken on the next state) of OptiDICE. The authors argue that directly adding the backward gradient may lead to its degeneration or cancellation if these two gradients conflict.
- To address this issue, the paper proposes a simple yet effective modification that projects the backward gradient onto the normal plane of the forward gradient, resulting in an orthogonal-gradient update, a novel learning rule for DICE-based methods.
- Through toy examples and extensive experiments on complex offline RL and IL tasks, the paper demonstrates that DICE-based methods using orthogonal-gradient updates achieve state-of-the-art performance and high robustness.

**Strengths:**

- Very interesting perspective on poorly performing OptiDICE and proposes a novel orthogonal update algorithm
- While the paper Is limited to DICE algorithm, the proposed orthogonal update seems to be able to be applied to other bootstrapping based deep RL methods.
- The paper is clearly presented and easy to follow.
- Experimental results are strong.

**Weaknesses:**

- The theoretical motivations for orthogonal gradient update do not seem to be sufficient. What we can know with the theoretical results presented are:
  - If we put right \eta, the orthogonal gradient can be no worse than semi-gradient
  - There is a possibility that orthogonal gradient can help feature co-adaptation
  - Based on these, O-DICE should perform on par with S-DICE on simple enough domains, and on complex domains, using DR3-like regularizations will make S-DICE to perform on par with O-DICE. Will there be additional source of better performance? Can we get better fixed point when orthogonal gradient with large \eta is adopted (large enough to make it different from S-DICE)?
- The Sikchi et al. (2023) trick seems to have a central role of the performance. details below.

**Questions:**

- Basically the orthogonal gradient technique used in this paper is not limited to DICE algorithms but we can adopt them on any deep RL algorithms. Can we get improvements on ordinary deep RL algorithms?

- As far as I know, if we use the semi-gradient update for OptiDICE, it diverges since the second term dominates the first term, and the second term only increases the V function according to the monotonic increasing shape of f^*. It seems the trick of Sikchi et al. (2023) is the trick that makes the algorithm to work. According to the objective in the paper, it seems the algorithm is sampling the arbitrary experience instead of initial state distribution, and it seems to be weighting it much more heavily. Is there any theoretical guarantee that the proposed objective function gives similar solution to what we can get with OptiDICE?

- OptiDICE actually tends to overestimate \nu, and similar to above, if S-DICE and O-DICE do not overestimate even without double Q trick, I believe that it should be related to the Sikchi et al. (2023) trick. is Sikchi et al. (2023) applied to OptiDICE in experiments?

- According to those reasons, I would like to see the results without the Sikchi et al. (2023) trick.

---

> ### Author Response · Authors · 2023-11-18
>
> We thank the reviewer for the effort engaged in the review phase and the constructive comments. Regarding the concerns and questions, we provide the detailed responses separately as follows.
>
> >The theoretical motivations for orthogonal gradient update do not seem to be sufficient. What we can know with the theoretical results presented are:
> If we put right \eta, the orthogonal gradient can be no worse than semi-gradient
> There is a possibility that orthogonal gradient can help feature co-adaptation
> Based on these, O-DICE should perform on par with S-DICE on simple enough domains, and on complex domains, using DR3-like regularizations will make S-DICE to perform on par with O-DICE. Will there be additional source of better performance? Can we get better fixed point when orthogonal gradient with large \eta is adopted (large enough to make it different from S-DICE)?
>
> 1. One important thing is that in Theorem 2, we proved that with appropriate $\eta$ (large enough), orthogonal-gradient update can make the overall loss objective monotonically decrease and get a better fixed point. This is not guaranteed by semi-gradient update since semi-gradient can lie in different half-plane than true-gradient. So orthogonal-gradient is actually better than semi-gradient from this perspective.
> 2. Although DR3-like regularization could make similar improvement on feature representations, introducing auxiliary loss can also change the original gradient flow of DICE methods and thus lose the convergence property above.
> 3. As shown in Theorem 2, 3, 4, we have additional better performance comes only by just utilizing projected backward gradient to enhance state-level constraint while elegantly retaining convergence properties as true-gradient.
>
> >Questions:
> Basically the orthogonal gradient technique used in this paper is not limited to DICE algorithms but we can adopt them on any deep RL algorithms. Can we get improvements on ordinary deep RL algorithms?
>
> We acknowledge that the orthogonal-gradient update could be applied to Bellman operator operations. However, a key factor contributing to the nice property brought by the orthogonal-gradient update in the DICE algorithm lies in the fact that the DICE algorithm does not minimize the traditional, **symmetric** MSBE (mean square Bellman loss), but instead minimizes an objective that incorporates a function $f^{\ast}$ that is **asymmetric**. As evident from Theorem 3, because in DICE $f^{*}$ is chosen to be monotonically increasing, $(f^\ast)'(x) \geq 0$ always holds so the RHS of Theorem 3 will less than 0.
> In other words, using orthogonal update in MSBE may not alleviate feature co-adaptation. Nevertheless, applying the orthogonal-gradient update to them may yield other intriguing properties, representing a potential avenue for future exploration.

---

> ### Author Response · Authors · 2023-11-18
>
> >As far as I know, if we use the semi-gradient update for OptiDICE, it diverges since the second term dominates the first term, and the second term only increases the V function according to the monotonic increasing shape of $f^*$. It seems the trick of Sikchi et al. (2023) is the trick that makes the algorithm to work. According to the objective in the paper, it seems the algorithm is sampling the arbitrary experience instead of initial state distribution, and it seems to be weighting it much more heavily. Is there any theoretical guarantee that the proposed objective function gives similar solution to what we can get with OptiDICE?
> >OptiDICE actually tends to overestimate \nu, and similar to above, if S-DICE and O-DICE do not overestimate even without double Q trick, I believe that it should be related to the Sikchi et al. (2023) trick. is Sikchi et al. (2023) applied to OptiDICE in experiments?
> >According to those reasons, I would like to see the results without the Sikchi et al. (2023) trick.
>
> First we want to clarify that the trick is not first proposed by Sikchi et al. (2023). It is a common trick used in DICE paper such as ValueDICE and IQLearn to increase the diversity of samples. Theoretically, using a wider initial state distribution of offline dataset does not affect the policy performance at the environment's initial state distribution as long as the function-approximator has enough capacity (in our case they are neural networks) since the policy will only learn to achieve near-optimal behaviors from a wider variety of states.
> Also, only O-DICE does not overestimate even without double Q trick, S-DICE still needs double Q trick to work. Suggesting the benefits come from using orthogonal update. The reason is that the gradient flow of $V$ can be reformulated as: (please refer to section 3.3)$$ (f^\ast)'(r+\gamma V(s') - V(s))\Big(\gamma\cdot\eta\nabla_\theta V(s') - \big(1 + \gamma\cdot \eta\frac{\nabla_\theta V(s)^\top \nabla_\theta V(s')}{\| \nabla_\theta V(s) \|^2}\big)  \nabla_\theta V(s) \Big) $$ Although $f^\ast$ is monotonic increasing, which would make $V(s_t)$ always increase or stay the same with the transition $(s_t, a_t, s_{t+1})$, $V(s_t)$ will decrease when we calculate gradient with the trasition $(s_{t-1}, a_{t-1}, s_t)$. Backward gradient will always make $V(s')$ decrease and thus remain undiverged.
>
> To better see the benefits, we compare O-DICE with OptiDICE without the Sikchi et al. (2023) trick. It can be seen that the benefit clearly comes from using orthogonal update rather than using $d^D$.
>
> |                 | O-DICE with $d_0$ | O-DICE with $d^{\mathcal{D}}$ | OptiDICE with $d_0$ | OptiDICE with $d^{\mathcal{D}}$ |
> | --------------- | ----------------- | ----------------------------- | ------------------- | ------------------------------- |
> | hopper-m        | 84.1              | 86.1                          |     49.9                |              46.4                   |
> | hopper-m-r      | 99.4              | 99.9                          |      14.3               |               20.0                  |
> | hopper-m-e      | 110.5             | 110.8                         |      58.1               |                51.3                 |
> | walker-m        | 82.2              | 84.9                          |        61.2             |            68.1                     |
> | walker-m-r      | 80.8              | 83.6                          |      17.8               |                17.9                 |
> | walker-m-e      | 110.6             | 110.8                         |       101.3              |                104.0                 |
> | halfcheetah-m   | 42.1              | 47.4                          |       42.0              |             45.8                    |
> | halfcheetah-m-r | 43.2              | 44.0                          |        32.5             |             31.7                    |
> | halfcheetah-m-e | 85.5              | 93.2                          |       53.0              |              59.7                   |

---

> > ### Comment · Reviewer_SNRk · 2023-11-20
> >
> > I do not agree that this trick is first used in Valuedice or IQLearn. I agree that they first used dataset distribution as initial distribution, and it is OK, since the optimal policy would not change. However, Sikchi et al. (2023) proposed to use $\lambda$ and $(1-\lambda)$ instead of $\gamma$, which results in different optimal solution.

---

> ### Author Response · Authors · 2023-11-23
>
> We believe we misunderstood the meaning of the Sikchi et al. (2023) trick. Initially, we interpreted it as using the dataset distribution instead of the initial distribution. As clarified in our paper, replacing $\alpha$ with $\lambda$ simply makes hyperparameter tuning easier. The optimal solution of $\alpha$-version is equivalent to that of $\lambda$-version with appropriate choice of $\alpha$. This is because both $\alpha$ and $\lambda$ function as hyperparameters to trade off the weights of the first linear term $V(s)$ and the second non-linear term $f^*(r+\gamma V(s') - V(s))$.
>
> In the $\alpha$-version, the ratio of the second gradient term's norm to the first gradient term's norm is given by:
> $$ \frac{ (f^{*})' ((r+\gamma V(s') - V(s)) / \alpha ) } {1-\gamma} \frac{|| \gamma \nabla V(s') - \nabla V(s) ||}{|| \nabla V(s) ||} $$
>
> For the $\lambda$-version, the ratio is expressed as:
> $$ \frac{ \lambda (f^{*})' (r+\gamma V(s') - V(s)) } {1-\lambda} \frac{|| \gamma \nabla V(s') - \nabla V(s) ||}{|| \nabla V(s) ||} $$
>
> Given $\lambda$, $\gamma$, $f$, and $V$, we can always find a corresponding $\alpha$ that equalizes these two ratios. Due to this equality, the full gradient of the $\alpha$-version aligns with that of the $\lambda$-version. With an appropriate learning rate, the solutions obtained using the $\alpha$-version and $\lambda$-version are nearly identical.
>
> To support our claim, we ran O-DICE using the $\alpha$-version (without employing Sikchi et al. (2023) trick), and the results closely match those presented in our paper." Experimental details remain consistent with those outlined in Appendix D, and the results are presented below.
>
> |                 | O-DICE with $\lambda$ (paper results) | O-DICE with $\alpha$ |
> | --------------- | --------------------- | -------------------- |
> | hopper-m        | 86.1                  | 83.2                 |
> | hopper-m-r      | 99.9                  | 100.1                |
> | hopper-m-e      | 110.8                 | 110.0                |
> | walker-m        | 84.9                  | 82.8                 |
> | walker-m-r      | 83.6                  | 82.1                 |
> | walker-m-e      | 110.8                 | 110.5                |
> | halfcheetah-m   | 47.4                  | 46.3                 |
> | halfcheetah-m-r | 44.0                  | 43.6                 |
> | halfcheetah-m-e | 93.2                  | 91.8                 |
> | antmaze-u       | 94.1                  | 94.0                 |
> | antmaze-u-d     | 79.5                  | 78.0                 |
> | antmaze-m-p     | 86.0                  | 85.5                 |
> | antmaze-m-d     | 82.7                  | 82.3                 |
> | antmaze-l-p     | 55.9                  |  51.0                    |
> | antmaze-l-d     | 54.0                  |   51.3                   |

---

### Official Review · Reviewer_8gZ1 · 2023-11-03

**Soundness:** 3 good
**Presentation:** 3 good
**Contribution:** 3 good
**Rating:** 8
**Confidence:** 4

**Summary:**

This paper explores DICE methods, an important area of research in offline RL. DICE-based methods impose behavior constraints at the state-action level, which is ideal for offline learning. They show that when learning the value function using true-gradient update, there are the forward gradient on the current state and the backward gradient on the next state. And they analyze that directly adding the backward gradient may cancel out its effect if the two gradients conflict. To address this, they propose a simple modification that projects the backward gradient onto the normal plane of the forward gradient, resulting in an orthogonal-gradient update. This new learning rule brings state-level behavior regularization. Through theoretical analyses, toy examples, and extensive experiments on complex offline RL and IL tasks, they demonstrate that DICE-based methods using orthogonal-gradient updates achieve good performance and robustness.

**Strengths:**

The advantages of this paper are as follows:
1. This paper discusses the relationship between true gradient and semi-gradient for (2) and establishes an analysis of the correlation between offline and online training using semi-gradient. This provides a new perspective on why (2) is difficult to train. As a result, the paper introduces the design of orthogonal-gradient, which is logically reasonable. Additionally, the paper presents the relationship between orthogonal-gradient and feature co-adaptation, making this design even more compelling.
2. This paper is written in a fluent manner with clear logic. The theoretical analysis is rigorous, and the experiments are abundant for comparison.

**Weaknesses:**

1. This paper claims that the only difference between itself and OptiDICE [1]  lies in whether to use orthogonal-gradient. However, from my understanding, there is a significant difference in the optimization objectives between OptiDICE and this paper. In [1], the corresponding optimization objective (11) includes not only the value function v but also the optimal density ratio w_v. Therefore, in terms of form, it is different from the optimization objective in this paper. As a result, the author's claim that they have found the mystery of why DICE-based methods are not practical is somewhat exaggerated.
2 . However, I believe that since (11) still involves v(s'), it should be possible to apply orthogonal-gradient when computing its gradient. Therefore, the author should compare the original OptiDICE algorithm with and without orthogonal-gradient to show that orthogonal-gradient has individual and significant gain.
3. Most RL algorithms involve Bellman operator operations, which require computing both forward and backward gradients. Therefore, orthogonal-gradient may not just be applicable to DICE-based methods, but also to most RL algorithms. This is much more important than improving DICE-based methods, and the author could consider this perspective.
4. From an experimental perspective, this method requires two key hyperparameters and needs different hyperparameters for different datasets, which is a disadvantage compared to other methods.

[1] Optidice: Offline policy optimization via stationary distribution correction estimation.

**Questions:**

1. The author claims that orthogonal-gradient can help consider state-level constraints, but there is no explicit explanation in the analysis as to why this constraint exists. Please provide a detailed explanation.
2. Is the $\epsilon$ in Theorem 4 different for different $s$ ? If so, I think Theorem 4 is trivial since we can easily find an state-wise $epsilon(s) $ to make  $V(s’+\epsilon)-V(s) $ flip sign.
3. The algorithmic complexity of O-DICE has not been analyzed.

---

> ### Author Response · Authors · 2023-11-18
>
> We thank the reviewer for the effort engaged in the review phase and the constructive comments. Regarding the concerns and questions, we provide the detailed responses separately as follows.
>
> >...from my understanding, there is a significant difference in the optimization objectives between OptiDICE and this paper. In OptiDICE, the corresponding optimization objective (11) includes not only the value function v but also the optimal density ratio w_v. Therefore, in terms of form, it is different from the optimization objective in this paper.
>
> If we understand correctly, you are referring to objective (13) in OptiDICE instead of (11). Actually (13) can be computed using only value function v because $\hat e_{\nu}$ is computed by v (see text under equation (8) in OptiDICE). OptiDICE additionally uses a neural network to learn $\hat e_{v}$ because they need to use that in the I-projection way to extract the policy: in I-projection, $\hat{e}_{\nu}$ is taking the role of critic and needs to provide gradient to $\pi$, so it needs to be a neural network. However, if we use the more common weighted BC method to extract the policy, then we do not need to learn one, we can just use the exact expression and it is unbiased.
> **In conclusion, the version of OptiDICE that uses weighted BC to extract the policy has the same optimization objective in our paper.**
>
> > Most RL algorithms involve Bellman operator operations, which require computing both forward and backward gradients. Therefore, orthogonal-gradient may not just be applicable to DICE-based methods, but also to most RL algorithms. This is much more important than improving DICE-based methods, and the author could consider this perspective.
>
> We acknowledge that the orthogonal-gradient update could be applied to Bellman operator operations. However, a key factor contributing to the nice property brought by the orthogonal-gradient update in the DICE algorithm lies in the fact that the DICE algorithm does not minimize the traditional, **symmetric** MSBE (mean square Bellman loss), but instead minimizes an objective that incorporates a function $f^{\ast}$ that is **asymmetric**. As evident from Theorem 3, because in DICE $f^{*}$ is chosen to be monotonically increasing, $(f^\ast)'(x) \geq 0$ always holds so the RHS of Theorem 3 will less than 0.
> In other words, using orthogonal update in MSBE may not alleviate feature co-adaptation. Nevertheless, applying the orthogonal-gradient update to them may yield other intriguing properties, representing a potential avenue for future exploration.
>
> >From an experimental perspective, this method requires two key hyperparameters and needs different hyperparameters for different datasets, which is a disadvantage compared to other methods.
>
> As we discussed in our limitation section, our method do need two hyperparameters, however, we find in practice that the tuning of $\eta$ is not sensitive.
> To validate this claim, we conduct ablation studies on $\eta$ using all D4RL datasets mentioned in Section 5.1. Experimental details remain consistent with those outlined in Appendix D, and the results are presented below. For MuJoCo datasets, a higher value of $\eta$ results in little or no performance drop, while a lower value of $\eta$ may induce instability and lead to small decrease in performance. In the case of AntMaze datasets, O-DICE also exhibits robustness to variations in $\eta$. O-DICE consistently outperforms the majority of offline RL methods across a wide range of $\eta$ values.
>
> |                 | $\eta=0.2$ | $\eta=0.4$ | $\eta=0.6$ | $\eta=0.8$ | $\eta=1.0$ |
> | --------------- | ---------- | ---------- | ---------- | ---------- | ---------- |
> | hopper-m        | 75.1       | 80.3       | 84.3       | 84.1       | 86.1       |
> | hopper-m-r      | 94.4       | 99.1       | 98.3       | 98.6       | 99.9       |
> | hopper-m-e      | 108.6      | 109.3      | 110.8      | 109.9      | 110.8      |
> | walker-m        | 84.9       | 84.2       | 83.2       | 79.5       | 79.9       |
> | walker-m-r      | 62.8       | 74.2       | 83.6       | 79.9       | 77.1       |
> | walker-m-e      | 110.8      | 110.9      | 110.2      | 110.7      | 110.9      |
> | halfcheetah-m   | 47.4       | 47.2       | 46.6       | 46.8       | 46.9       |
> | halfcheetah-m-r | 44.0       | 43.6       | 42.4       | 43.0       | 42.9       |
> | halfcheetah-m-e | 93.2       | 92.8       | 91.1       | 92.1       | 91.2       |
> | antmaze-u       | 92.0       | 90.6       | 92.4       | 92.3       | 94.1       |
> | antmaze-u-d     | 68.7       | 70.3       | 72.0       | 74.0       | 79.5       |
> | antmaze-m-p     | 86.0       | 78.3       | 70.0       | 61.9       | 55.1       |
> | antmaze-m-d     | 82.7       | 75.0       | 68.3       | 57.5       | 49.9       |
> | antmaze-l-p     | 36.1       | 44.0       | 52.7       | 55.9       | 32.0       |
> | antmaze-l-d     | 38.7       | 46.3       | 52.0       | 54.0       | 30.5       |

---

> ### Author Response · Authors · 2023-11-18
>
> >I think Theorem 4 is trivial since we can easily find an state-wise $\epsilon$ to flip sign.
> >The author claims that orthogonal-gradient can help consider state-level constraints, but there is no explicit explanation in the analysis as to why this constraint exists. Please provide a detailed explanation.
>
> Theorem 4 is not trying to prove that $\epsilon$ will always exist. Rather, we aim to demonstrate that, with a smaller value of $\Psi_{\theta}(s, s')$, the inequality condition $\Psi_{\theta}(s, s')>M-C\cdot \| \epsilon \|^2$ requires $\epsilon$ to have a larger norm in order to change the sign of $V(s'+\epsilon) - V(s)$. In other words, the existence of $\epsilon$ capable of causing a sign flip in $V(s'+\epsilon) - V(s)$ must satisfy the condition $\Psi_{\theta}(s, s')>M-C\cdot \| \epsilon \|^2$. This condition can be reformulated as $\| \epsilon \|^2 > \frac{M - \Psi_{\theta}(s, s')}{C}$, indicating that it becomes harder for $V(s'+\epsilon) - V(s)$ to flip sign with a smaller $\Psi_{\theta}(s, s')$.
>
> If the sign (or value) of $V (s') − V (s)$ can be easily flipped by some noise $\epsilon$, first the policy extraction part during training will be affected (as two similar s′ will have totally different BC weight), then the optimal policy induced from V is likely to be misled by OOD states encountered during evaluation and take wrong actions.
> For example, for transition $<s,a_1, s'_1>$ in the offline dataset, suppose $s'_1$ is more near the boundary of dataset support than $s$ and the true value of $V(s'_1)$ is smaller than $V(s)$, suppose another transition $<s,a_2, s'_1+\epsilon>$ is also in the dataset. Then the policy will be misled by $a_2$ because the learned $V$ will be affected by the small noise $\epsilon$, according to Theorem 4. If it takes $a_2$ then it will go to OOD states more easily. An ideal choice is do not believe either $a_1$ and $a_2$ because they are equally bad, by using orthogonal gradient update we can better do that.
> >The algorithmic complexity of O-DICE has not been analyzed.
>
> As evident in the pseudocode, O-DICE only needs to exclusively compute the backward gradient and perform gradient projection. The algorithmic complexity of these two steps is equivalent to calculating the forward gradient, a step common to all other DICE algorithms. Consequently, O-DICE's algorithmic complexity is no heavier than that of other DICE algorithms.

---

> > ### Comment · Reviewer_8gZ1 · 2023-11-20
> >
> > Thanks for answering my questions! You have addressed most of  my concerns and I've raised my score to 8.

---

> ### Author Response · Authors · 2023-11-23
>
> Thank you for your valuable feedback and for considering our revisions. We appreciate your positive reassessment of our work. Your insights have greatly contributed to the improvement of our paper.

---

### Official Review · Reviewer_LpDw · 2023-11-07

**Soundness:** 3 good
**Presentation:** 4 excellent
**Contribution:** 4 excellent
**Rating:** 8
**Confidence:** 3

**Summary:**

The paper proposes Orthogonal-DICE algorithm, that incorporates the V-DICE algorithm with orthogonal-gradient update. The gap between EQL and OptiDICE is analyzed and theoretical proofs are given. Experimental results show that the proposed method achieve better performance than many state-of-the-art methods.

**Strengths:**

see questions

**Weaknesses:**

see questions

**Questions:**

The paper is well-written and easy to follow. The high-level idea is clear with interesting motivation. Theoretical analysis shows the reason harm of backward gradient. The experimental result is also impressive and shows the difference in gradients clearly. I have one question about a special case when gradients are computed
1. How do you calculate the orthogonal gradient when the angle between backward and forward gradients is more than 90 degrees? Especially when it is 180 degrees?

---

> ### Author Response · Authors · 2023-11-18
>
> >How do you calculate the orthogonal gradient when the angle between backward and forward gradients is more than 90 degrees? Especially when it is 180 degrees?
>
> We thank the reviewer for the effort engaged in the review phase and the constructive comments. Regarding the concerns, we provide the detailed responses separately as follows.
>
> When the angle between the backward and forward gradients exceeds 90 degrees, we continue to employ the following equation to calculate the orthogonal gradient: $$(f^\ast)'(r+\gamma V(s') - V(s))\Big(\gamma\cdot\eta\big( \nabla_\theta V(s') - \frac{\nabla_\theta V(s)^\top \nabla_\theta V(s')}{\| \nabla_\theta V(s) \|^2}\nabla_\theta V(s) \big)  - \nabla_\theta V(s) \Big)$$
>
> Note that in this case, the parallel component of the backward gradient lies in the same direction as the forward gradient, it may introduce instability to the forward gradient due to this uncontrollable interference. Therefore, we still need to exclude the parallel component and utilize only the orthogonal part of the backward gradient.
>
> When the angle is 180 degrees, the true gradient is equal to the forward gradient. Using the equation above, we can calculate that the orthogonal gradient is also equal to the true/forward gradient because the orthogonal term is 0.

---

### Meta-Review · Area_Chair_GG4W · 2023-12-05

**Metareview:**

(a) this paper proposes an analysis of why the DiCE based methods for offline RL are often less effective. They argue this is because of a cancelling of the forward and backward gradient in DiCE. They have a nice discussion of this phenomenon and then essentially propose a gradient projection method to deconflict gradients. They show compelling results on many offline RL settings.

(b) nice simple insight, clear motivation and simple solution. This is a nice paper! Also backed by both theoretical results and practical implementation.

(c) My main gripe with this paper is that they completely ignore and don't cite PCGrad[1] , which essentially does the same orthogonal gradient approach, although for multi-task RL. The paper still has merit and is a good one, but I think it does reduce the novelty of the work. The other reviewers also brought up the fact that this could be applied to *any* deep RL method, and in fact methods like PCGrad do apply it to other problems. Despite this, the paper has made valuable contributions and should be accepted.

(d) a discussion of PCGrad and other orthogonal gradient methods. Addition of new results requested by reviewers in the rebuttal.

[1] Gradient surgery for multi-task learning: Yu, Tianhe and Kumar, Saurabh and Gupta, Abhishek and Levine, Sergey and Hausman, Karol and Finn, Chelsea, NeurIPS 2020

**Justification For Why Not Higher Score:**

The novelty of the method is not as high since previous methods like PCGrad. Otherwise the method is a nice contribution and shows strong results.

**Justification For Why Not Lower Score:**

Reviewers are in consensus this is a great paper. It has strong empirical and theoretical results.

---

### Decision · Program_Chairs · 2024-01-16

Accept (spotlight)